# Structural basis of reactivation of oncogenic p53 mutants by a small molecule: methylene quinuclidinone (MQ)

Oksana Degtjarik [1], Dmitrij Golovenko [1,4], Yael Diskin-Posner [2], Lars Abrahmsén[3], Haim Rozenberg [1✉] & Zippora Shakked [1✉]

In response to genotoxic stress, the tumor suppressor p53 acts as a transcription factor by regulating the expression of genes critical for cancer prevention. Mutations in the gene encoding p53 are associated with cancer development. PRIMA-1 and eprenetapopt (APR-246/PRIMA-1[MET]) are small molecules that are converted into the biologically active compound, methylene quinuclidinone (MQ), shown to reactivate mutant p53 by binding covalently to cysteine residues. Here, we investigate the structural basis of mutant p53 reactivation by MQ based on a series of high-resolution crystal structures of cancer-related and wild-type p53 core domains bound to MQ in their free state and in complexes with their DNA response elements. Our data demonstrate that MQ binds to several cysteine residues located at the surface of the core domain. The structures reveal a large diversity in MQ interaction modes that stabilize p53 and its complexes with DNA, leading to a common global effect that is pertinent to the restoration of non-functional p53 proteins.

[1] Department of Chemical and Structural Biology, Weizmann Institute of Science, 76100 Rehovot, Israel. [2] Department of Chemical Research Support, Weizmann Institute of Science, 76100 Rehovot, Israel. [3] Aprea Therapeutics AB, 17165 Solna, Sweden. [4] Present address: University of Massachusetts Medical School, RNA Therapeutics Institute, Worcester, MA 01605, USA. ✉email: haim.rozenberg@weizmann.ac.il; zippi.shakked@weizmann.ac.il

The human p53 tumor suppressor is a 393-residue protein that in response to cellular stress, acts as a transcription factor by binding as a tetramer to a wide range of DNA response elements, activating various events including DNA repair, cell-cycle arrest, senescence or apoptosis[1,2].

Mutations in the p53 gene were observed in more than half of human cancers. These mutations are mostly a consequence of single amino-acid substitutions that result not only in the loss of wild-type p53 function, but occasionally also in endowing new gain-of-function activities such as enhanced resistance to chemotherapy or genetic instability[3–6]. Around 97% of the tumorigenic mutations are found in the DNA-binding core domain[7–9], spanning nearly 200 residues (94–293) and comprising a large β-sandwich and a DNA binding surface that encompasses a loop-sheet-helix motif and two loops supported by a zinc ion (e.g.[10,11]).

Mutations of arginine at positions 175, 248, 249, 273, 282, and of glycine at position 245 are referred as hotspots due to their high frequency in various types of cancer, comprising nearly 30% of the somatic mutations in the core domain[7–9]. These mutations, located at or near the protein-DNA interface[10,11], lead to p53 inactivation by loss of direct p53-DNA interactions, and/or by causing conformational changes in the protein, resulting in lowering its stability[12–15]. The first type includes mutations at positions 248 and 273 and referred as DNA-contact mutations. The second type includes mutations at positions: 175, 245, 249 and 282, and referred as structural mutations. The stability and folding states of the DNA-contact mutants are comparable to that of wild-type p53 whereas those of the structural mutants vary depending on the position and identity of the specific amino-acid replacement[15].

Arginine at position 273 is one of the most frequently altered residues in p53 among somatic mutations[8]. It has been shown to be predominantly mutated in cancer to histidine and cysteine (47.5% and 39.1%, respectively)[8], resulting in largely reduced DNA binding activity[15] and hence loss of function. The effects of these mutations have been demonstrated by the crystal structures of the corresponding mutants, R273H and R273C[16,17]. Among the p53 structural mutants, R282W and R175H were shown to be largely destabilized and unfolded[15]. The other structural mutants, G245S and R249S retain the overall 3-D structures, showing local structural changes near the specific mutations[17,18].

Several small molecules have been investigated as potential drugs to restore wild-type function to mutant p53 (Reviewed in[19–22]). The most clinically advanced, eprenetapopt (APR-246/PRIMA-1MET, https://www.clinicaltrials.gov) is a close analogue of PRIMA-1 (p53-Reactivation and Induction of Massive Apoptosis-1), which was discovered in a screen for compounds reactivating mutant p53[23]. PRIMA-1 was shown to induce apoptosis in human tumor cells via restoration of transcriptional activation, as well as restoring the active conformation, and sequence-specific DNA binding to both structural and DNA-contact p53 mutants[23]. Similarly, it has been shown that APR-246 exhibits strong anti-leukemia activity in ALL cells derived from *TP53*mut patients, targeting non-functional p53 including structural and DNA-contact mutants[24]. These properties of PRIMA-1 and APR-246 have been confirmed in many studies performed on human tumor cells and xenograft tumor systems expressing various p53 mutants[25–30]. In addition, activation of the p53 pathway has been shown in clinical studies: the first-in-human study[31] and in a study on the treatment of patients with myelodysplastic syndrome by eprenetapopt and azacytidine combination[32]. Results from the later study did not suggest any specific p53 mutation associated with better response.

Both PRIMA-1 and APR-246 are prodrugs converted into the biologically active compound methylene quinuclidinone (MQ)[33].

MQ is a Michael acceptor, a very reactive molecule preferentially binding reversibly to soft nucleophiles, including cysteine thiol groups of the p53 core domain[33]. Whereas some reversible interactions of MQ do not lead to observable effects, others contribute to triggering p53-independent cell death, for example by inhibiting thioredoxin and glutaredoxin antioxidant systems leading to increased levels of reactive oxygen species[34–36]. MQ also binds and depletes glutathione and, not surprisingly, high intracellular levels of glutathione reduce the effect of treatment[37,38]. Based on thermal stability measurements and cell-based assays, replacing several Cys residues in the p53 core domain by Ala residues, it was proposed that C277 was essential for MQ-mediated stabilization of the core domains of wild-type p53, the structural mutant R175H and the DNA-contact mutant R273H. However, both C124 and C277 were required for MQ-mediated functional restoration of R175H in tumor cells[39]. In a previous study it was shown that the replacement of C124 by alanine abolished p53 reactivation of R175H mutant by PRIMA-1[40].

To uncover the structural basis of rescuing p53 mutants by MQ we investigated by X-ray crystallography high-resolution structures of the core domains of mutant and wild-type p53 bound to MQ in the absence and/or the presence of their DNA response elements. These studies reveal diverse binding modes of MQ to specific cysteine residues that stabilize the proteins and their interaction with DNA. On the basis of the structural data in conjunction with previous biochemical and cell-based data on MQ binding to p53, we propose a mechanistic explanation for the functional rescue of oncogenic p53 as a tumor suppressor.

## Results

**High-resolution structural data on MQ bound to mutant and wild-type p53.** In the current study we present the binding modes of MQ to p53 based on high-resolution crystal structures of mutant and wild-type (wt) p53 core domains in their free state and/or bound to DNA. Systematic crystallization and MQ incorporation experiments of the various molecular systems led to crystal structures of MQ bound to the p53 DNA-binding core domain (referred as p53DBD; see Methods and Supplementary Tables 1–9). Two enantiomers may be formed when the Michael acceptor MQ binds to a thiol group (see Supplementary Note 1). This reaction is reversible, and hence the formation and selection of either enantiomer at a specific Cys residue varies, depending on stabilizing interactions as described below. The wt p53 core domain includes ten Cys residues where three of them (C176, C238 and C242) interact with a structural Zn ion, supporting the DNA binding surface[10,11]. Most of the other wild-type Cys residues including C124, C182, C229, C275, C277 and the hotspot mutation product C273 are located at or close to the protein surface and are accessible to MQ.

The studied oncogenic p53DBD mutants include two DNA-contact mutants, R273H and R273C, and a structural mutant, R282W[7,8]. The replacement of Arg273 by His or by Cys residues eliminates key stabilizing interactions between p53 and DNA, causing a large decrease in binding affinity[15,16], whereas the replacement of Arg282 by a Trp residue results in a significant conformational change leading to protein destabilization[12,17]. Here we present structural data on MQ binding to Cys residues in the core domain of the following systems: R273H and R273C mutants, DNA complexes of R273H, R282W, the double mutant R273C/S240R (incorporating a second-site suppressor mutation) and wild-type p53 (see Methods). The modified residues in each structure are listed in Table 1.

The structures of the MQ-modified p53 proteins (mutant and wt) reveal a range of MQ-Cys conjugates and their interaction modes.

**Table 1 Bound MQ observed in structures of free p53 core domains and p53-DNA complexes.**

| Structure/Residue | C124 | C182 | C229 | C273 | C275 | C277 | K101 |
|---|---|---|---|---|---|---|---|
| Free p53<br>R273H-MQ (I)[a] | **A**[g]<br>**B**[g] | **B** MQr (0.89)<br>**C**[g] | A MQs (0.71)<br>B MQr (0.27)<br>+ MQs (0.48)[b]<br>C MQr (0.79)<br>D MQs (0.60) | N.A. | A MQs (0.75)<br>**B**[g]<br><br>D MQr (0.44)<br>+ MQs (0.37)[b] | **C**[g]<br>D MQs (0.59) | |
| R273H-MQ (II)[a] | | **B** MQr (0.80)<br>**C** ACT (0.70)[c] | | N.A. | | **C** MQr (0.85) | |
| R273C-MQ (I)[a] | | **B** MQr (0.84) | A MQs (0.76)<br>B MQs (0.86)<br>C MQr (0.67)<br>**D**[g] | A MQr (0.58)<br>+ MQr (0.31)[b]<br>B MQs (0.79)<br>C MQr (0.52)<br>+ MQs (0.35)[b]<br>D MQr (0.87) | **A**[g]<br>**B**[g]<br>C MQs (0.64) | B MQr (0.90) | |
| R273C-MQ (II)[a] | | **B** MQr (0.85)<br>**C** MQr (0.75) | | A MQr (0.80)<br>B MQs (0.75)<br>C MQs (0.85)<br>D MQr (0.70) | | B MQs (0.80) | |
| p53-DNA complexes<br>R273H-DNA-MQ[d] | **A/B**[g] | | **A/B** MQs (0.76) | N.A. | | **A/B** MQr (0.61) | |
| R282W-DNA-MQ[e] | A MQs (0.59) | | A MQs (0.74) | | | A MQr (0.50)<br>+MQs (0.50)[b] | |
| R273C/S240R-DNA-MQ[e] | A MQr (0.86) | | A MQr (0.35)<br>+ MQs (0.29)[b] | | | A MQr (0.59) | |
| wt-DNA-MQ (I)[e] | A MQr (0.88) | | A MQr (0.35)<br>+ MQs (0.45)[b] | | | A MQr (0.96) | |
| wt-DNA-MQ (II)[e] | A MQs (0.78) | | A MQs (0.75) | | | A MQr (0.89) | |
| wt-DNA-MQ (III)[e] | A MQs (1.0) | | A MQs (1.0) | | | A MQr (1.0) | |
| wt-DNA-MQ (P1)[a,f] | **B**[g]<br><br>**D**[g] | | A MQs (0.74)<br>**B**[g]<br>C MQs (0.61)<br>D MQr (0.89) | | | A MQs (0.79)<br>B MQs (0.88)<br>C MQs (0.77)<br>D MQs (0.70) | A MQr (0.88) |

N.A. stands for "Not Applicable".
[a]Four independent p53DBD monomers in the asymmetric unit of the crystal (**A**, **B**, **C**, **D**).
[b]Bound MQ observed in two alternative enantiomers or orientations.
[c]Acetate ion (ACT) observed at a location used by bound MQ in other monomers.
[d]Two p53DBD monomers related by non-crystallographic symmetry in the asymmetric unit (**A/B**).
[e]A single p53DBD monomer in the asymmetric unit (**A**).
[f]A single MQ bound to lysine (K101) was observed. MQ-K101 was trapped in the crystal of wt-DNA-MQ (P1), between two neighboring p53-DNA tetramers (see text for details).
[g]Un-modeled bulk of electron density with high probability of being covalently-bound MQ.

The MQ-modified structures are compared with the previously reported MQ-free structures[11,16,41] and the currently determined new structures of R273H and R282W complexes with DNA.

**MQ bound to R273H and R273C mutants**. The crystal structures of R273H and R273C incorporating covalently-bound MQ were determined at a resolution range of 1.71–2.05 Å from crystals with four independent p53DBD molecules (Supplementary Tables 6, 7). The individual MQ-modified molecules share a similar fold with that of the previously reported R273H and R273C structures[16] (RMSD range: 0.7–1.1 Å based on all atoms). The specific MQ conjugates appear to be dependent on the procedure used to obtain the crystals (Supplementary Table 1). In the crystal structures obtained by the soaking procedure referred as R273H-MQ (I) and R273C-MQ (I), bound MQ has been identified at residues: C182, C229, C275 and C277 in both structures, as well as at C273 in R273C-MQ (I) (Table 1). However, in the crystal structures obtained by co-crystallization with MQ, referred as R273H-MQ (II) and R273C-MQ (II), bound MQ has been identified only at residues: C182 and C277 in both mutants as well as at C273 in R273C-MQ (II) (Table 1).

Cysteine at position 182 is located on the flexible surface loop L2, making it the most solvent-accessible residue in the structures of the two DNA-contact mutants[16], and in other p53 structures[42,43]. In the current crystal structures of MQ-bound p53 mutants, only two of the four monomers (B and C) show MQ-C182 conjugates (Table 1). This selective modification appears to result from the exposure of the specific C182 side chains to the solvent and the stabilizing interactions of the bound MQ with neighboring molecules in the crystal, shown in Fig. 1. In the other two monomers (A and D), as well as in the structures of the unmodified mutants[16], C182 side chains are facing toward the

protein core, and are less accessible to MQ (Supplementary Fig. 1). The current MQ-C182 conjugates display a similar orientation associated with an extended chain conformation, stabilized via stacking interaction with a tryptophan (W146), hydrogen bonds and/or CH···O interactions with R110 from a neighboring molecule in the crystal. Because the observed modifications of C182 residues are supported by intermolecular interactions depending on the arrangement of p53 molecules in the crystal and do not appear in the p53-DNA-MQ structures described below, the role of MQ-C182 on p53 stabilization and hence reactivation is not clear.

MQ-C229 conjugates are observed solely in the crystal structures R273H-MQ (I) and R273C-MQ (I), obtained by the soaking procedure (Table 1, Supplementary Table 1). MQ-C229 conjugates are partly shielded by the large S7/S8 loop and residues from S3 and S8 strands shown in Fig. 2. Both enantiomers (MQs and MQr) were identified in each mutant, interacting differently with the surrounding regions via hydrogen bonds, CH···O and van der Waals (VdW) interactions. The MQs conjugates form well-directed hydrogen bonds via their carbonyl oxygen with the backbone amide of D228, as well as water-mediated hydrogen bonds via their tertiary amine with the side chain of T231 and the backbone carbonyl oxygen of C229 (Fig. 2a, c). In addition, electrostatic interactions made by parallel CH···O contacts are present between MQs and S7/S8 loop (Fig. 2a, c). Fewer interactions are shown by the MQr conjugates, resulting from the alternative chirality and orientation of MQ (Fig. 2b, d). Thus, in addition to intramolecular hydrogen bonds within the S7/S8 loop observed also in the absence of MQ[16], the conformation of S7/S8 loop is further supported by several MQ-mediated interactions.

Well-defined MQ-C275 conjugates are present in two p53 monomers of R273H-MQ (I) and one monomer of R273C-MQ

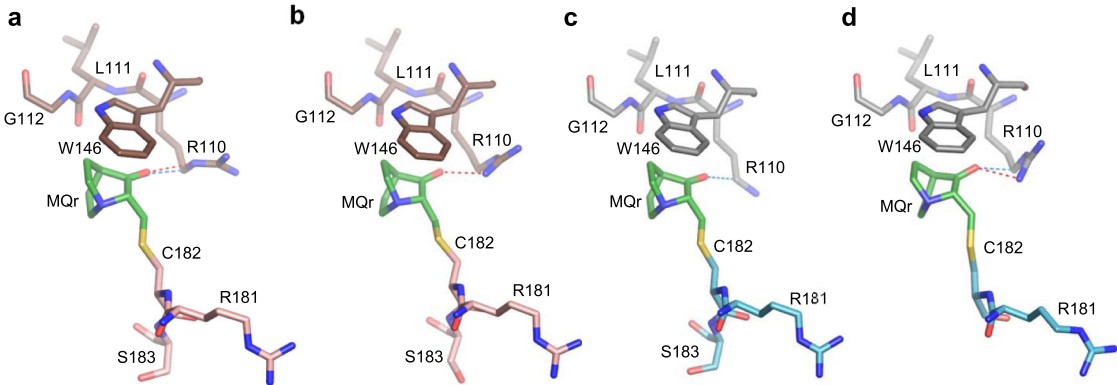

**Fig. 1 MQ bound to C182 in R273H and R273C mutants.** The MQ-C182 conjugates and interactions with neighboring molecules are from the following crystal structures. **a** R273H-MQ (I), monomer B and symmetry-related monomer C. **b** R273H-MQ (II), monomer B and symmetry-related monomer C. **c** R273C-MQ (I), monomer B and symmetry-related monomer C. **d** R273C-MQ (II), monomer C and symmetry-related monomer B. All MQ-C182 conjugates interact with neighboring molecules via stacking interactions with W146, hydrogen bonds and CH···O interactions with R110 side chains. The missing side chains or atoms are disordered and not well defined. The structures are in stick representation with the following color codes: C atoms are shown in green for MQ, pink and brown for the MQ-modified and neighboring monomers, respectively, of R273H-MQ (I, II), and light blue and grey for the corresponding monomers of R273C-MQ (I, II). N, O and S atoms are in blue, red and yellow, respectively. Hydrogen bonds and CH···O interactions are indicated by red and blue dotted lines, respectively.

(I) (Table 1). Bound MQs is observed in monomer A, and both enantiomers (with partial occupancy) in monomer D of R273H-MQ (I) (Fig. 3a, b, respectively). These conjugates form several stabilizing interactions, including hydrogen bonds made by the carbonyl oxygen and the tertiary amine of MQ with N239 and with S241 side chains, and with the backbone amide of A276. Also observed are water-mediated interactions between the amino group of N239 and the backbones of L137 and C275 (Fig. 3a). Additional MQ-mediated contacts include CH···O and VdW interactions with the side chains of Y107 and S106 from neighboring monomers, respectively. These interactions stabilize MQ-C275 conjugates in monomers A and D of R273H-MQ (I). Unlike the other MQ-modified Cys residues, the current MQ-C275 conjugates are not compatible with DNA binding because of their predicted close proximity to DNA (Supplementary Fig. 2a). MQs bound to C275 in monomer C of R273C-MQ (I) forms CH···O interactions with MQr bound to C273, and with the side chain of D281 (Fig. 3c). This conjugate is also incompatible with DNA binding (Supplementary Fig. 2b).

C277 is the second most accessible cysteine in structures of p53 proteins in the absence of DNA[42,43]. C277 is located at the protein-DNA interface and involved in base-specific interactions[10,11]. MQ-C277 conjugates were identified in four monomers, one in each of the four crystal structures (Table 1), shown in Fig. 4a–d. Stabilizing interactions include water-mediated hydrogen bonds between the polar edges (O and N atoms) of MQs or MQr with residues from L1 loop (121–122) and residues from H2 helix (279–280), as well as with neighboring molecules. Also observed are MQ-mediated intra- and intermolecular CH···O or VdW interactions (Fig. 4a–c).

C273 residue is unique among the p53 cysteins by being a result of a DNA-contact hotspot mutation common in human cancer. Interestingly, this residue is extensively modified in all eight monomers of the two R273C-MQ structures (Table 1). C273 is located in a shallow depression at the protein surface surrounded by residues from several regions of the protein, and hence providing an attractive target for the reactivation of R273C by a small drug molecule. Five of the MQ-C273 conjugates are shown in Fig. 5a–e. In all monomers, a pivotal hydrogen bond is

present between the tertiary amine of MQ and the amino group of K132, being further supported by the negatively charged E285 side chain. Other stabilizing interactions include direct and water-mediated hydrogen bonds and CH···O interactions of MQ with neighboring residues from other regions (S240, R248, I251, E271 and V272). A common water molecule is anchoring several residues to form a stabilizing network of hydrogen bonds and CH···O interactions (Fig. 5a–d). These MQ-C273 conjugates are likely compatible with DNA binding as indicated by comparing the structure of an R273C-MQ monomer with that of wild-type p53 monomers interacting with DNA (Supplementary Fig. 3a).

**MQ bound to p53 proteins in their complexes with DNA.** All crystals were obtained by soaking p53-DNA crystals into MQ solutions (Supplementary Table 1), and analyzed at a resolution range of 1.32–2.0 Å (Supplementary Tables 2–9). The MQ-modified structures include complexes of two hotspot mutants: the DNA contact mutant R273H and the structural mutant R282W, referred as R273H-DNA-MQ and R282W-DNA-MQ, respectively (Table 1). Also analyzed here, are the crystal structures of the corresponding MQ-free complexes referred as R273H-DNA and R282W-DNA (I, II). Crystals of R273H-DNA were obtained in the presence of DNA and MQ in the crystallization solution, but without traces of bound MQ (Supplementary Table 1). The role of MQ in this process is not clear. Crystals of R282W-DNA were obtained with and without MQ (Supplementary Table 1) due to the relative high binding affinity of the complex[44]. In addition, we present the structures of MQ bound to DNA complexes of the rescued double mutant R273C/S240R, referred as R273C/S240R-DNA-MQ, and two types of MQ-modified wt complexes referred as wt-DNA-MQ (I-III) and wt-DNA-MQ (P1) (Table 1). The corresponding MQ-free structures were determined previously[11,16,41].

The DNA response elements of the current complexes are made of 12-mer or 21-mer DNA oligomers, incorporating one or two consensus sequences (GGGCATGCCC), respectively, used previously[11,16,41]. In each case, two decameric half-sites interact with two p53 dimers to form a p53-DNA tetramer. The two half-

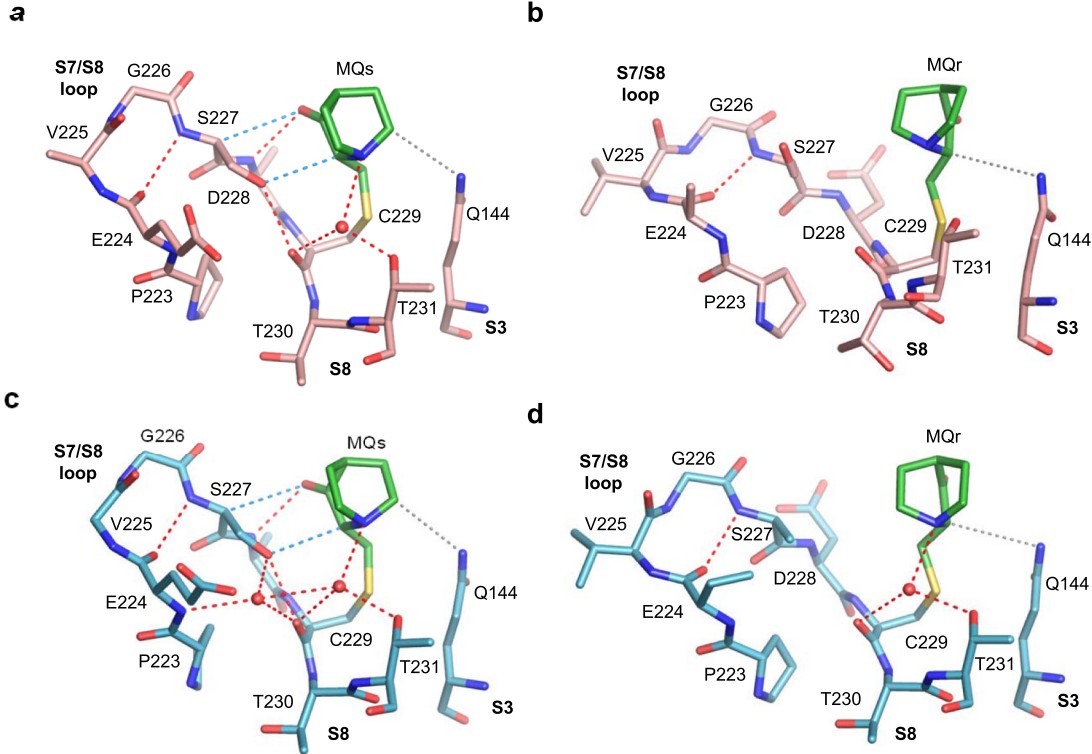

**Fig. 2 MQ bound to C229 in R273H and R273C mutants.** The MQ-C229 conjugates and intramolecular interactions are from the following crystal structures. **a** R273H-MQ (I), monomer A. **b** R273H-MQ (I), monomer C. **c** R273C-MQ (I), monomer A. **d** R273C-MQ (I), monomer C. MQ-C229 is located at the C-terminus of S7/S8 loop. Direct and water-mediated hydrogen bonds, and CH···O interactions are indicated by red and blue dotted lines, respectively. Van der Waals (VdW) interactions are in grey dotted lines. Water molecules shown as red spheres. Other structure representations and color codes are as in Fig. 1.

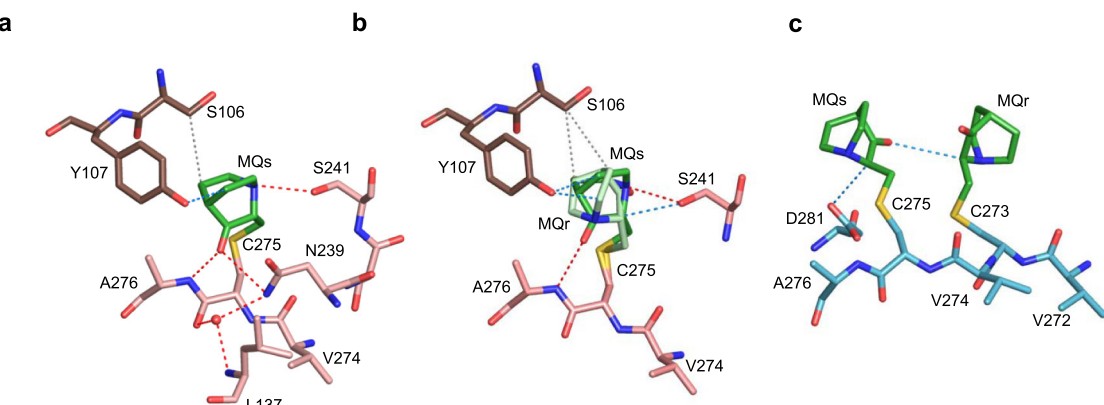

**Fig. 3 MQ bound to C275 in R273H and R273C mutants.** The MQ-C275 conjugates and intra- and intermolecular interactions are from the following crystal structures. **a** R273H-MQ (I), monomer A and symmetry-related monomer B. **b** R273H-MQ (I), monomer D and symmetry-related monomer C. **c** R273C-MQ (I), monomer C. Two alternative Cys-bound MQ enantiomers are shown in (**b**) (MQr in light green and MQs in dark green). MQ-C275 in the structure of R273C-MQ (I) is close to the nearby MQ-C273 conjugate. Other structure representations and color codes are as in Figs. 1, 2.

sites are contiguous via either covalent linkage or base-pair stacking interactions within a continuous 20 bp DNA helix, except for wt-DNA-MQ (P1) where the two half-sites are separated by two stacked base pairs simulating a 22-bp helix (schematic DNA helices of the various complexes are in Supplementary Table 10).

The MQ-modified complexes of p53-DNA show clearly defined MQ conjugates at three Cys residues: C124, C229 and C277 (Table 1). Superposition views of each of the tetrameric complexes R273H-DNA, R282W-DNA, R273C/S240R-DNA and

wt-DNA (I), prior to and after binding to MQ, are shown in Fig. 6a–d, highlighting the similarity between them in each p53-DNA complex. Close-up views of the MQ-modified Cys residues within the core domain structure, from each of the p53-DNA tetramers shown in Fig. 6, are displayed in Supplementary Fig. 4.

A detailed description of the various MQ conjugates given below reveals a high diversity of MQ binding and interaction modes in the various p53-DNA complexes and their mutual stabilization.

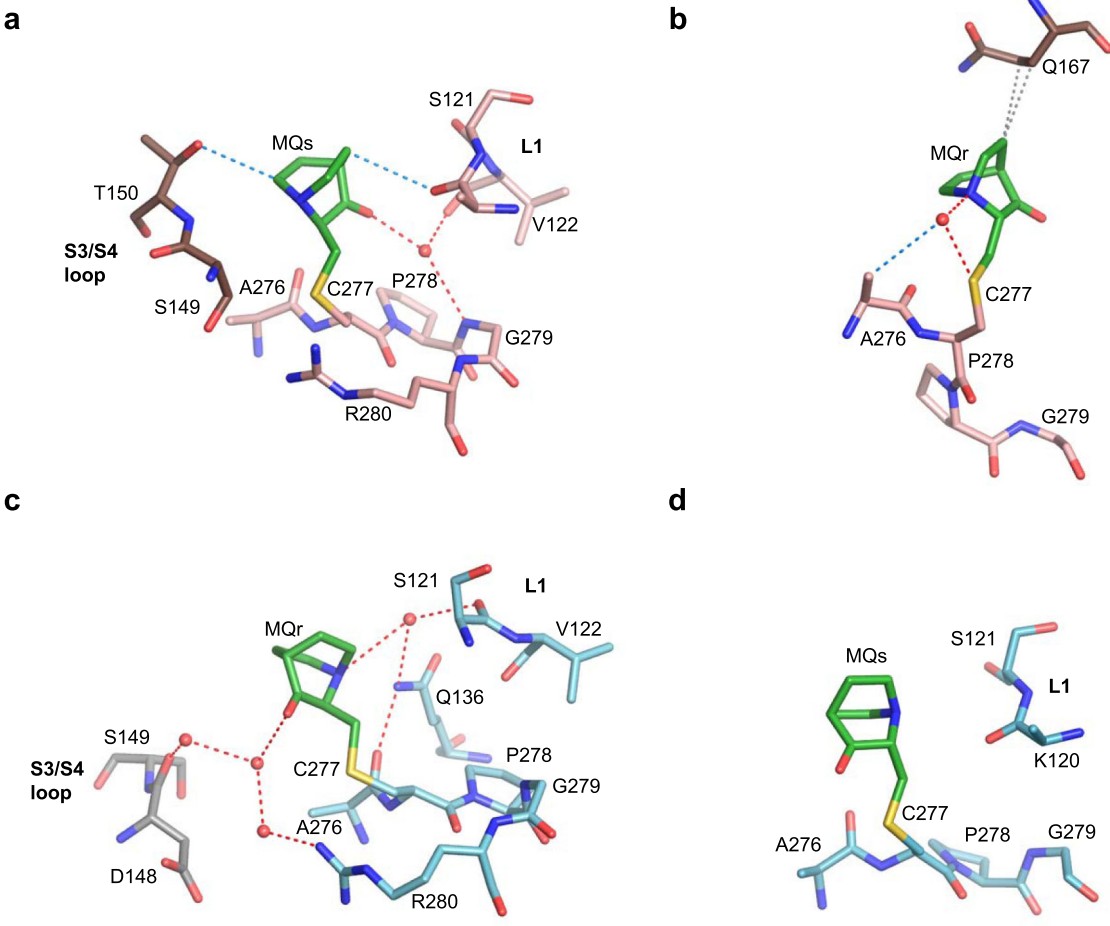

**Fig. 4 MQ bound to C277 in R273H and R273C mutants.** The MQ-C277 conjugates and interactions with the molecular environment are from the following crystal structures. **a** R273H-MQ (I), monomer D and monomer C. **b** R273H-MQ (II), monomer C and symmetry-related monomer D. **c** R273C-MQ (I), monomer B and symmetry-related monomer A. **d** R273C-MQ (II), monomer B. The MQ-C277 conjugates show a large variance in MQ orientation and interactions. Structure representations and color codes are as in Figs. 1–3.

MQ-C124 is well defined in five structures: R282W-DNA-MQ, R273C/S240R-DNA-MQ, and wt-DNA-MQ (I, II, III). In addition, residual bulks of electron density close to the sulfur atom of C124 are observed in the other two structures, R273H-DNA-MQ and wt-DNA-MQ (P1), but too weak for unambiguous MQ modeling (Table 1).

C124 is located at the protein surface in a shallow depression made by the β-strand S3 (residues 141–143) and the N- and C-termini of the L1 loop (residues 114–116 and 122–124). The orientation of MQ-C124 varies among the five structures, and hence its specific interactions with the nearest neighbors as shown in Fig. 7a–e. In addition to interactions with L1 and S3 residues of the same p53 molecule, each MQ conjugate forms stabilizing contacts with an adjacent p53 dimer via its polar L2 residues (S166, Q167), and occasionally with the overhung G12 nucleotides (Fig. 7b, e). These contacts include direct and water-mediated hydrogen bonds as well as CH⋯O and VdW interactions.

The L1 loop (residues 113–124) which supports MQ binding to C124 and involved in stabilizing MQ-C277 conjugates in the DNA-free mutants (Fig. 4) and p53-DNA complexes (described below), is considered to be one of the most flexible elements in

the p53 core domain[45]. L1 has been shown to be occasionally disordered in previous p53-DNA complexes[16,18], and significant disorder in L1 is present in several of the current p53-DNA-MQ structures, and thus only partially defined (Supplementary Table 11). By contrast, L1 is well ordered in the structures of R273H and R273C mutants[16] and the corresponding MQ-modified structures studied here, showing a common L1 conformation (RMSD range of 0.3–0.9 Å based on all atoms). Whereas the free space available for accommodating MQ-C124 conjugates is similar between the modified and unmodified p53-DNA structures (with the exception of R282W-DNA-MQ), the space available for MQ-C124 conjugates in the DNA-free structures of R273H-MQ and R273C-MQ is restricted relative to that of the DNA-bound proteins (illustrated in Supplementary Fig. 5).

The structure of R282W-DNA-MQ displays a large change in MQ-C124 orientation and environment in comparison to the other MQ-modified complexes (Fig. 7a–e and Supplementary Fig. 5a–f). In this structure, MQ-C124 conjugate adopts an extended backbone conformation, attaining stabilizing interactions with P142 of S3 and F113 of L1 (Fig. 7a). A comparison among the structures of wt-DNA, R282W-DNA and R282W-DNA-MQ

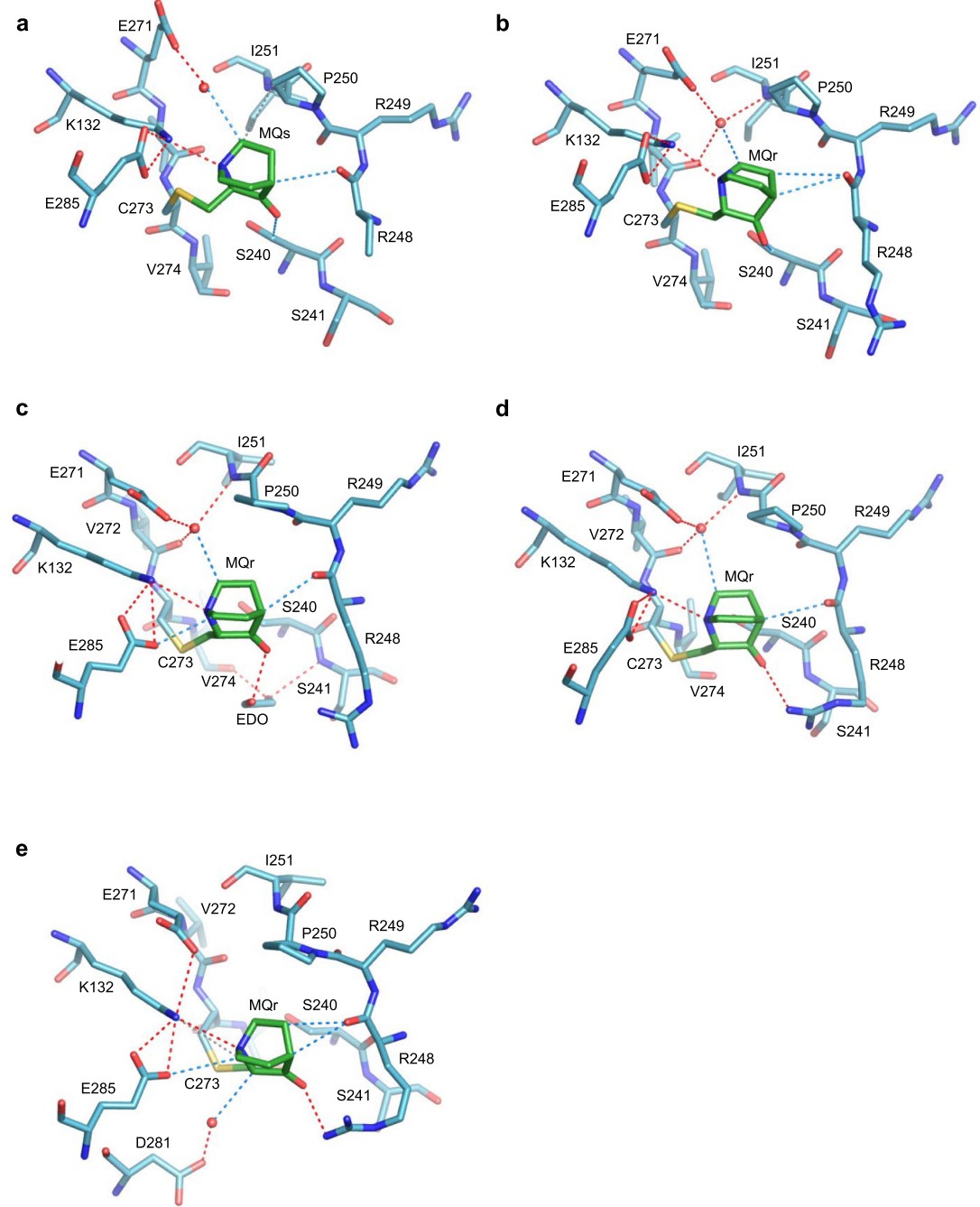

**Fig. 5 MQ bound to C273 in R273C mutants.** The MQ-C273 conjugates and intramolecular interactions are from the following crystal structures. **a** R273C-MQ (I), monomer B. **b** R273C-MQ (I), monomer D. **c** R273C-MQ (II), monomer A (EDO refers to ethylene diol). **d** R273C-MQ (II), monomer B. **e** R273C-MQ (II), monomer C. All MQ-C273 conjugates are similarly located in a shallow depression at the p53 surface and supported by polar interactions with the surrounding residues. Structure representations and color codes are as in Figs. 1–3.

demonstrates the local effects of the structural mutation W282 and of MQ binding on the corresponding R282W-DNA complexes shown in Supplementary Fig. 6a–d. The L1 loop in the two R282W-DNA structures is well defined showing two alternative conformations, whereas this loop in R282W-DNA-MQ is largely disrupted (Supplementary Fig. 6b–d).

Notwithstanding the observed variance in chirality and orientation of the MQ-C124 species, they all support the interface between p53 dimers of each p53 tetramer (referred as *AB* and *CD*

in Fig. 6). These interactions in turn interfere with the release of MQ molecules from C124 residues.

MQ bound to C229 is observed at high occupancy in all p53-DNA-MQ structures (Table 1). The MQ-C229 conjugates display a large range of orientations and hence interactions with p53, by virtue of the relatively large space around MQ, shown in Fig. 8. These conjugates are in close proximity to residues from the S7/S8 loop, and S3 and S8 strands observed also in the modified DNA-free mutants (shown in Fig. 2 at a slightly different view).

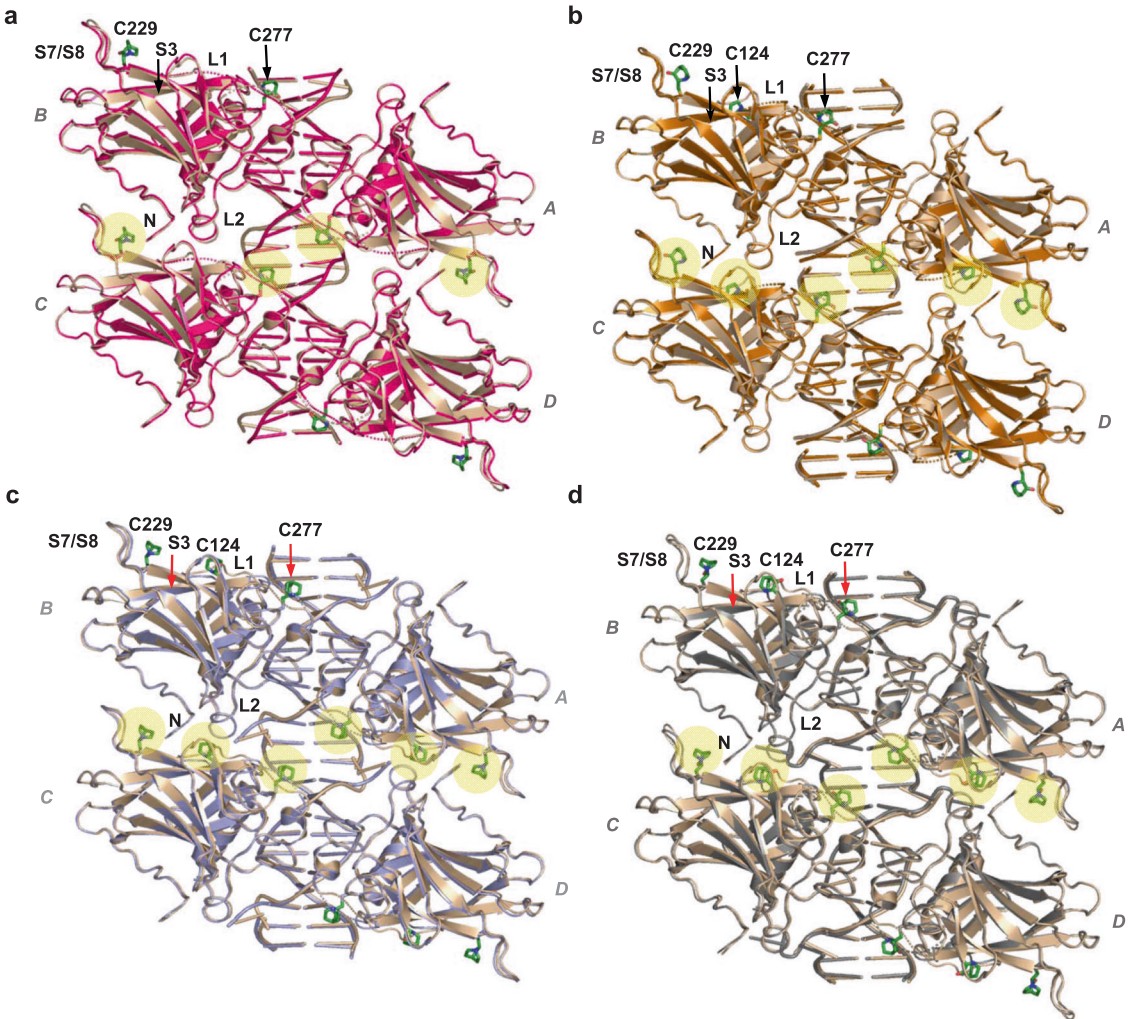

**Fig. 6 MQ bound to C124, C229 and C277 in mutant and wild type p53-DNA tetramers.** The structures of tetrameric p53-DNA-MQ complexes superposed on their MQ-free counterparts are the following. **a** R273H-DNA-MQ (magenta) and R237H-DNA (light brown). **b** R282W-DNA-MQ (copper) and R282W-DNA (light brown). **c** R273C/S240R-DNA-MQ (light blue) and R273C/S240R-DNA (PDB ID: 4IBV, light brown). **d** wt-DNA-MQ (I) (grey) and wt-DNA (PDB ID: 3IGL, light brown). The MQ-Cys conjugates are present in all four monomers due to crystal symmetry. The tetrameric p53-DNA structures are in cartoon representation. The MQ conjugates are in stick representation (green) and highlighted by yellow spheres at the horizontal region between the dimers (*AB* and *CD*). The MQ-modified Cys residues and adjacent protein regions (L1, L2, S7/S8, and S3) of each tetramer are labeled at the top left-handed p53 monomer. The presented DNA helices are made of 20 base pairs in (**a**), and 22 base pairs in (**b–d**) (the DNA constructs are shown in Supplementary Table 10). Close-up views of the MQ-Cys conjugates within the corresponding core domains are shown in Supplementary Fig. 4.

In some of the p53 molecules, the carbonyl oxygen of MQ is pointing towards the S7/S8 loop forming a hydrogen bond with the backbone amide of D228 (Fig. 8b, g, h). In the other structures, where the carbonyl oxygen is pointing away from S7/S8, several water-mediated hydrogen bonds are made by the tertiary amine of MQ with the S7/S8 backbone, and the hydroxyl group of the nearby T231 residue from the S8 strand (Fig. 8a, c–f). Water-mediated interactions between the distant MQ carbonyl oxygen and S7/S8 are also observed (Fig. 8a, e, f). Other stabilizing contacts between MQ-C229 and p53 include CH⋯O interactions with S7/S8, and VdW interactions with the adjacent Q144 and W146 residues from the S3 strand.

Stabilizing contacts between MQ-C229 and p53 residues are also observed in the structures of R273H-MQ and R273C-MQ described above. Comparisons between these regions in the free and the DNA-bound molecules, show enhanced supporting interactions in the later. The S7/S8 loop incorporates an intra-loop hydrogen bond between the backbone carbonyl and the amide groups of amino acids E224 and S227, respectively. A

second intra-loop hydrogen bond is observed between the hydroxyl of S227 and the backbone carbonyl of C229. The second hydrogen bond is further supported by MQ in the p53-DNA structures, but to a lesser extent in the DNA-free structures (Compare Figs. 2 and 8).

In the p53-DNA complexes where the two DNA half-sites are contiguous and hence the two dimers are parallel to each other, MQ-C229 conjugates are located at the inter-dimer interface (Fig. 6). Amino acids from the S7/S8 loop (E224 and G226) interact with amino acids from the adjacent dimer: R267 from the S10 strand, and S99, K101 from the N-terminus (shown in Fig. 8a–f). These interactions include salt bridges and hydrogen bonds. In particular, optimally-oriented intermolecular bidentate salt bridges between the negatively and positively charged residues are observed in the structure of R282W-DNA-MQ attained by a small tilt of the S7/S8 loop leading also to VdW interaction of MQ with K101 from the neighboring dimer (Fig. 8b). Such interactions at the inter-dimer interface are dependent on the structural stability of the S7/S8 loop. Hence,

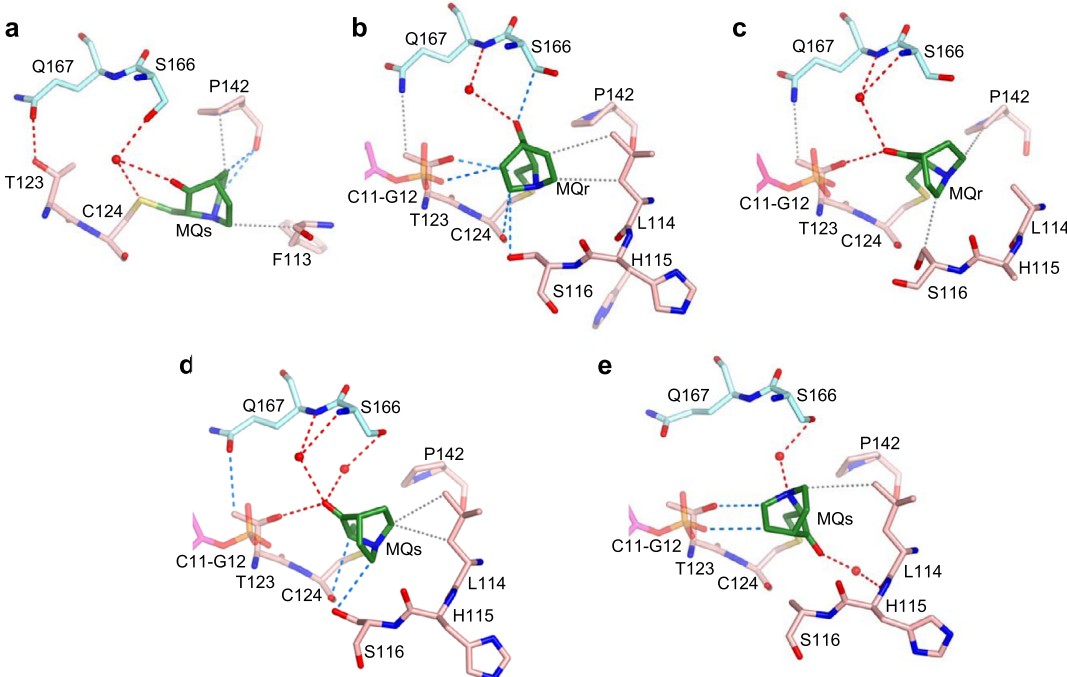

**Fig. 7 MQ bound to C124 in p53-DNA complexes.** The MQ-C124 conjugates and associated intra- and intermolecular interactions are based on the following structures. **a** R282W-DNA-MQ. **b** R273C/S240R-DNA-MQ. **c** wt-DNA-MQ (I). **d** wt-DNA-MQ (II). **e** wt-DNA-MQ (III). MQ-C124 is located at the C-terminus of the flexible L1 loop (residues 113–124). R282W-DNA-MQ differs largely from the other four structures by the orientation and position of the bound MQ (see text for details). The structures are in stick representation with the following color codes: C atoms shown in green for MQ, pink for the MQ-modified p53 monomers and light blue for the neighboring monomers. N, O, S and P atoms are in blue, red, yellow and orange, respectively. Hydrogen bonds, CH···O and VdW interactions shown in red, blue and grey dotted lines, respectively, and water molecules shown as red spheres.

MQ-C229 contributes to the structural integrity of the p53-DNA tetramer by supporting this loop.

Inter-dimer interactions by the S7/S8 loop do not exist in the wt-DNA (P1) complex and its MQ-modified structure (Fig. 8g, h), because the two p53 dimers are rotated relative to each other owing to the two base-pair spacer between the half-sites, resulting in different inter-dimer interactions[11]. The single MQ-Lys conjugate (MQr-K101) observed in the current study has been identified in a single monomer (A) of this structure (Table 1). Here, K101 is not involved in inter-dimer interactions shown by the other structures, thereby allowing its free access to MQ. The MQ-K101 conjugate is stabilized extensively through intra- and intermolecular interactions (see Supplementary Fig. 7). The occurrence of this conjugate is surprising because the amino nitrogen is much less reactive than a thiol sulfur in the Michael Addition. This conjugate appears to result from the large accessibility of this amino group and its massive stabilization by residues of two monomers from neighboring p53-DNA tetramers in the crystal. Hence, it is unlikely to form under physiological conditions.

MQ-C277 conjugates were observed in all structures of mutant and wt p53-DNA complexes. The MQ conjugates are located at the protein-DNA interface and surrounded by L1 loop, residues adjacent to C277 and the DNA major groove (displayed in Fig. 9a–e). C277 by itself is a DNA-binding amino acid, interacting via its thiol group with the first base of the consensus pyrimidine triplet of each half-site shown previously in crystals of p53-DNA complexes[10,11,41,46–48]. This interaction is retained between the sulfur of MQ-C277 and the first cytosine base of the CCC triplet of each DNA half-site in all modified complexes (assigned as C8′ in Fig. 9a and C9′ in Fig. 9b–e; base numbering in Supplementary Table 10).

New hydrogen bonds with DNA bases, mediated by MQ-C277, are observed only in R273H-DNA-MQ and R282W-DNA-MQ (Fig. 9a, b). In these structures, the MQ carbonyl oxygen is oriented toward the DNA, interacting with the second base of each CCC triplet (assigned as C9′ in Fig. 9a and C10′ in Fig. 9b). This orientation would lead to a steric clash between the other side of the bound MQ and L1 loop at S121 and adjacent residues, thus enforcing conformational rearrangements of L1, resulting in significant disorder of this region in both structures. This effect is indicated by missing residues and partially resolved residues in L1, unlike the corresponding unmodified structures displaying well-defined L1 loops (Supplementary Table 11). Additional stabilizing interactions at the protein-DNA interface of the two mutants, include CH···O and CH···N interactions made by MQ with the first base of the GGG triplet of each DNA half-site (assigned as G11 in Fig. 9a and G2 in Fig. 9b), and with the carbonyl backbone of residue A276.

The presence of MQ-C277 in R273H-DNA-MQ instigates a major alternative conformation in the DNA backbones of dinucleotides C9-C10 and its symmetry-related C9′-C10′. As a result, the phosphate groups from the two symmetry-related nucleotides, C10 and C10′, are shifted by 3.7 Å towards the DNA helix center, allowing for CH···O interactions with the two symmetry-related MQr-C277 conjugates from two p53 monomers (assigned as A and C in Fig. 6a). Close-up views of these interactions and other stabilizing contacts at the central p53-DNA interface are shown in Fig. 9a and Supplementary Fig. 8. This type of interactions driven by an alternative DNA backbone conformation does not exist in the other structures because of a different DNA architecture (Supplementary Table 10).

In the other MQ-modified DNA complexes, the carbonyl oxygen of MQ is oriented away from the DNA and toward the L1

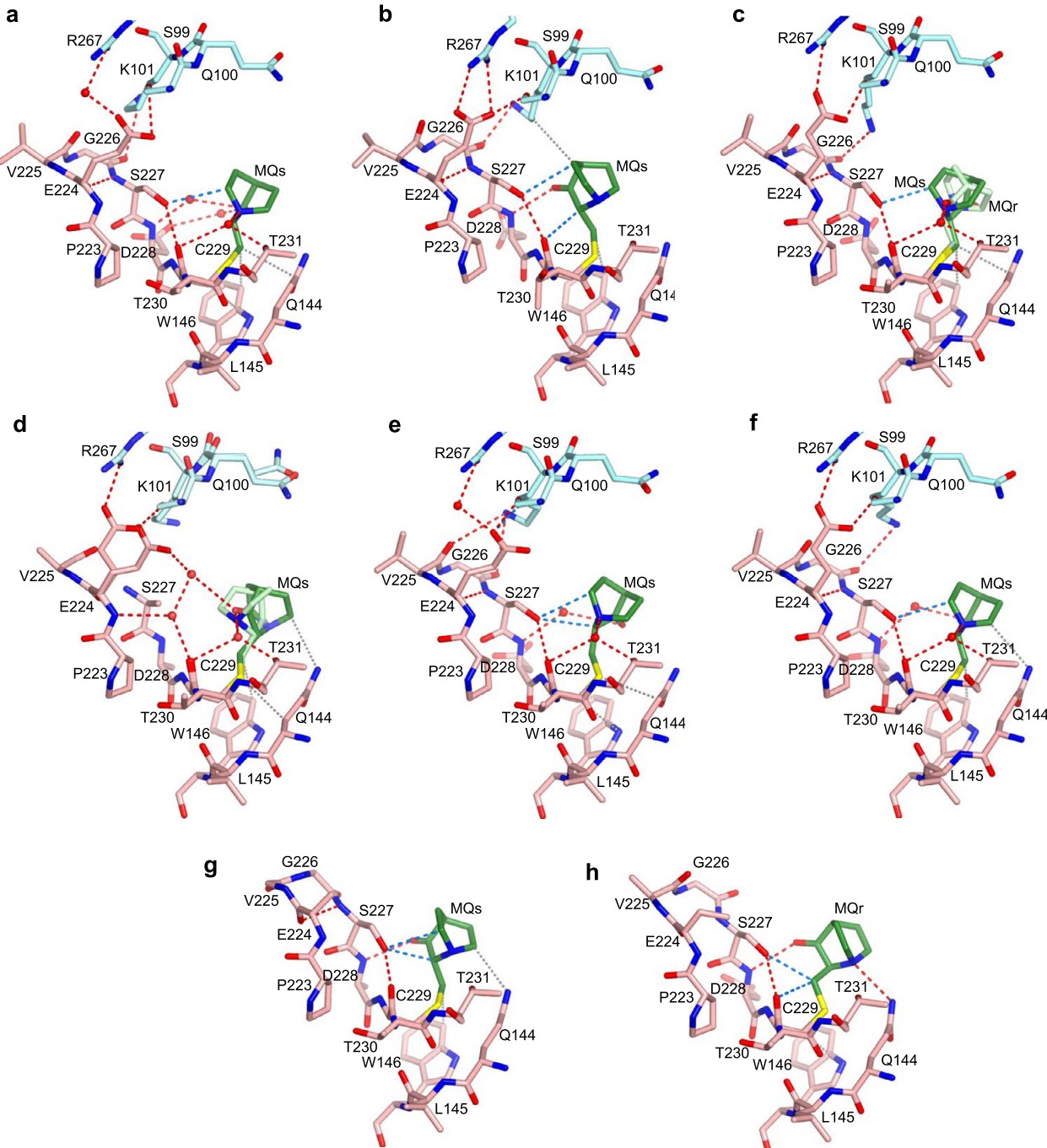

**Fig. 8 MQ bound to C229 in p53-DNA complexes.** The MQ-C229 conjugates and the stabilizing interactions are based on the following structures. **a** R273H-DNA-MQ. **b** R282W-DNA-MQ. **c** R273C/S240R-DNA-MQ. **d** wt-DNA-MQ (I). **e** wt-DNA-MQ (II). **f** wt-DNA-MQ (III). **g** wt-DNA-MQ (P1), monomer A. **h** wt-DNA-MQ (P1), monomer D. MQ-C229, located at the C-terminus of the S7/S8 loop, supports the structure of this loop via a network of stabilizing interactions, thereby contributing to the stability of the inter-dimer interface (see text for details). Significant local disorder at the turn of S7/S8 (residues V225, G226, S227) is observed in wt-DNA-MQ (I) shown in **d**, associated with the presence of two alternative Cys-bound MQ enantiomers (MQr in light green and MQs in dark green), and two alternative conformations of E224 and of residues S99-Q100 from the neighboring p53 dimer. One E224 side chain forms a salt bridge with R267 and a hydrogen bond with S99 from the neighboring dimer, shown also by the other structures, whereas the other E224 side chain forms a hydrogen bond via a water network with the carbonyl oxygen of MQs bound to C229. The second E224 conformation is also present in several other structures, but not involved in stabilizing interactions (not shown here for clarity). Unlike the structures shown in (**a**–**f**), the inter-dimer interactions mediated by S7/S8 loop are not present in the wt-DNA-MQ (P1) structure (**g, h**), because the two p53 dimers of this structure are rotated relative to each other, thereby leading to different inter-dimer contacts which are not affected by MQ-C229 (see text for details). Structure representations and color codes are as in Fig. 7.

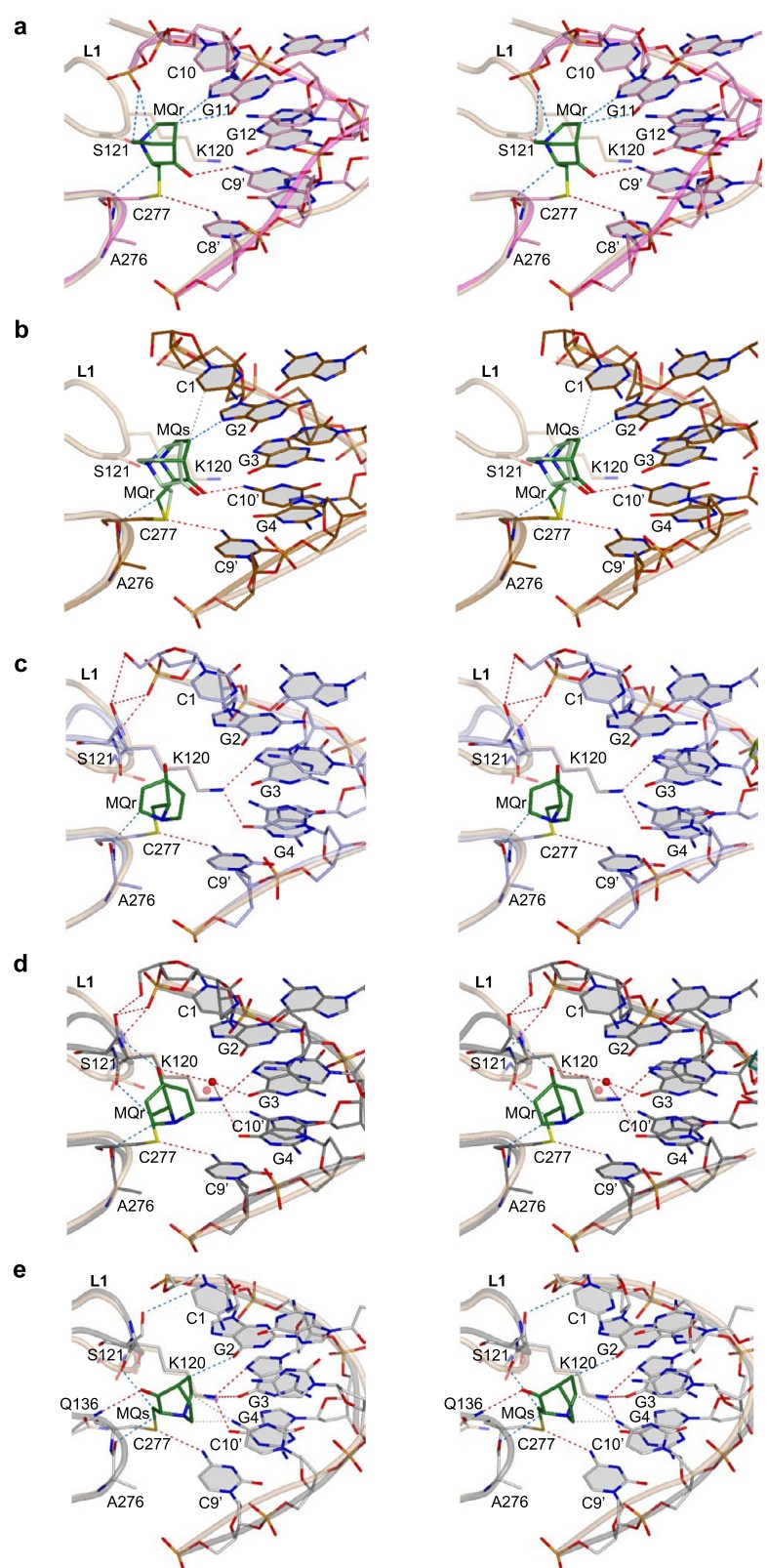

loop. A local conformational change in L1, relative to that of the MQ-free complexes, leads to different stabilizing contacts between S121 and the backbone of the 5′-end nucleotides without affecting the nearby K120 interactions with G3 and G4 bases (Fig. 9c–e). In these structures, no direct MQ-mediated hydrogen bonds with the DNA bases are observed, except for a water-mediated hydrogen bond between the MQ carbonyl oxygen and

G3 base observed in the three structures of wt-DNA-MQ (I, II, III) represented by structure (I) in Fig. 9d. In the other type of complex, wt-DNA-MQ (P1), a significantly different DNA helix (see Supplementary Table 10) leads to alternative interactions mediated by MQs (Table 1, Fig. 9e). Here, the stabilizing interactions made by MQ, include a hydrogen bond via its carbonyl oxygen with Q136 residue, and CH···O interactions with

**Fig. 9 MQ bound to C277 in p53-DNA complexes.** Divergent stereo views of MQ-C277 conjugates and the stabilizing interactions at the protein-DNA interface, superposed on their MQ-free counterparts, are based on the following structures. **a** R273H-DNA-MQ (magenta) and R273H-DNA (light brown). **b** R282W-DNA-MQ (copper) and R282W-DNA (I) (light brown). **c** R273C/S240R-DNA-MQ (light blue) and R273C/S240R-DNA (PDB ID: 4IBV, light brown). **d** wt-DNA-MQ (I) (dark grey) and wt-DNA (PDB ID: 3IGL, light brown). **e** wt-DNA-MQ (P1) (grey) and wt-DNA (PDB ID: 2AC0, monomer C, light brown). The structures are in stick representation for MQ, DNA and specific p53 regions at the protein-DNA interface, combined with cartoon representations of the corresponding p53 and DNA backbones. Only cartoon representations of the DNA backbones of the MQ-free complexes are displayed (light brown). In addition to the regular p53-DNA hydrogen bonds of the various complexes in this region (via K120 and C277), the modified protein-DNA interfaces are stabilized by MQ-mediated hydrogen bonds, CH···O and VdW interactions indicated by red, blue and grey dotted lines, respectively (see text for details). The water molecule (shown as a red sphere in (**d**) between the carbonyl group of MQ and the amino group of G3 base is common to all three wt-DNA-MQ structures, and also present in the unmodified complex (shown as a pink sphere in (**d**). The MQ-C277 conjugate affects the conformation of the nearby L1 loop. A significant disorder of L1 is observed in R273-DNA-MQ and R282W-DNA-MQ structures (**a, b**), and hence only L1 loops of the unmodified complexes are shown. Numbering of the DNA bases differs between R273H-DNA-MQ (**a**) and the other structures (**b–e**), depending on the DNA constructs shown in Supplementary Table 10. For clarity, in this figure the base numbers of one of the two DNA strands are primed.

the carbonyl group of the G2 base and the backbone carbonyls of A276 and K120. Other supporting contacts are VdW interactions between MQ and a cytosine base (C10′ in Fig. 9e).

Our previous studies of p53-DNA complexes have demonstrated a diversity in the local DNA shape, depending on the geometry of A/T base pairs at the centers of the DNA half-sites: the regular Watson-Crick (WC) or the unusual Hoogsteen (HG) base pairs. The HG geometry was first observed in consensus p53 response elements with contiguous DNA half-sites incorporating central A/T base pairs[41], and later in natural p53 response elements[48]. Structural data on p53-DNA complexes incorporating His273 and Cys273 mutations demonstrated that the unusual HG geometry is dependent on Arg273 interactions with the DNA backbone between the A/T base pairs of continuous half-sites and thus, p53-DNA structures incorporating DNA-contact mutations or non-contiguous DNA half-sites, are likely to adopt the regular WC geometry[11,16]. This pattern is observed also in the current structures (Supplementary Table 10).

## Discussion

To decipher the reactivation mechanism of p53 mutants by MQ, we conducted high-resolution structural studies on both DNA-contact and structural mutants and their complexes with DNA. In the current DNA-contact mutants, R273H and R273C, critical hydrogen bonds between Arg side chains and DNA phosphate groups are lost by the replacement of an Arg residue by a His or a Cys residue[16]. Whereas the stability of such mutants is comparable to that of wt p53, their DNA-binding affinity is marginal[44]. By contrast, the stability of the current structural mutant, R282W, is largely reduced by the replacement of an Arg residue by a Trp residue, yet exhibiting significant DNA-binding affinity[44]. The instability of p53 caused by Trp282 mutation has been suggested to result from the disruption of an extended intramolecular hydrogen-bond network, based on comparison between the structures of the thermostable p53 core domain and its corresponding R282W mutant[17]. Without secondary stabilizing mutations, such disruptive effects likely propagate to adjacent regions in the R282W mutant studied here, interfering with crystallization. Crystal structures of the free DNA-contact mutants of the core domain, R273H and R273C, were reported previously[16]. However, crystals of the structural R282W mutant were achieved here only in complexes with DNA, yielding high-resolution crystal structures. A possible explanation is that the DNA response element acts as a "chaperone" by stabilizing p53 proteins as reported previously[12]. The MQ-modified cysteines and their stabilizing effects are summarized in Tables 1, 2.

MQ bound to Cys residues in the various crystal structures was achieved by two methods: soaking of preformed crystals in MQ solutions and co-crystallization experiments in the presence of MQ. Covalent binding of MQ to thiol groups is a reversible reaction[38], and hence the equilibrium is likely shifting towards the bound state when favorable interactions are made between MQ and p53 or with both p53 and DNA, resulting in their mutual stabilization. The Cys residues are located at various protein sites of different molecular surroundings. Hence, the modification levels of the specific Cys residues are dependent on both their accessibility to MQ in solution and/or the crystalline state, and on the stability of the MQ conjugates.

Soaking crystals of R273H and R273C mutants in MQ solutions led to well-defined MQ conjugates at five residues: C182, C229, C275, C277 and C273, whereas co-crystallization of the protein mutants with MQ yielded crystals incorporating MQ conjugates at three residues: C182, C277 and C273.

In the soaking procedure, the small molecules diffuse into a crystal made of ordered protein molecules in three dimensions and interact reversibly with accessible Cys residues. In solution and prior to crystallization, it is likely that MQ binds reversibly to additional Cys residues shown previously by other alkylating agents[42]. However, some MQ conjugates occurring in solution might interfere with the 3-D arrangement of the protein molecules in the growing crystal or are not retained in the absence of stabilizing interactions. C273 is unique among the modified Cys residues as being an oncogenic mutation. It is noteworthy that MQ bound to C273 is present at high occupancy levels in all p53 monomers of the two types of structures, R273C-MQ (I, II), obtained by the two different procedures (Table 1). Each MQ bound to C273 is located in a depression at the protein surface and not interfering with crystallization, as well as showing extensive stabilization by intramolecular interactions with the surrounding p53 residues. Such environmental conditions are required for kinetically stable MQ-Cys conjugates in both types of crystals.

Interestingly, the common MQ-modified residues, C182 and C277 observed in all crystal structures of the current p53 mutants in their free state, display the largest solvent accessibility[42,43]. These residues were also reported previously as being the cysteines preferentially modified by alkylating agents[43,49], and that specific modification of C182 and C277 increased p53 stability without compromising DNA binding[49]. In the later study, a crystal structure obtained by soaking crystals of the oncogenic p53 mutant Y220C[17] with an alkylating agent, showed that the small molecule is covalently bound to C182, and indicated a second binding site by a bulk of electron density close to C277[49]. The Y220C mutant was shown previously to provide an excellent model for the development of drugs to rescue proteins incorporating cavity-creating cancer mutations[50].

Co-crystallization of the current p53 mutants and of wt p53 with both DNA and MQ yielded crystals of the proteins bound to

**Table 2 Effects and significance of MQ-modified Cys residues.**

| Residue | Number of MQ-Cys conjugates[a] | | Location | Effects and significance |
|---|---|---|---|---|
| | DNA-free p53 | DNA-bound p53 | | |
| C124 | 0 | 5 | C-terminus of L1 loop | MQ-C124 is not resolved in the DNA-free structures because of limited accessibility of C124 to MQ. MQ-C124 is observed in p53-DNA tetramers, forming stabilizing interactions, and is highly significant in reinforcing the inter-dimer interface. |
| C182 | 5 | 0 | L2 loop | MQ-C182 is observed in the DNA-free structures, depending on an outward L2 conformation, supported by MQ interactions with an adjacent p53 monomer in the crystal structures and hence its effect on p53 stability is not clear. |
| C229 | 7 | 9 | C-terminus of S7/S8 loop | MQ-C229 is observed in both DNA-free and DNA-bound structures, forming stabilizing interactions. Enhanced MQ-mediated stabilization is observed in the p53-DNA tetramers, reinforcing the inter-dimer interface. |
| C273 | 8 | N/A[b] | C-terminus of S10 strand | MQ-C273 is observed in all p53 monomers of the MQ-modified R273C structures, showing similar patterns of extensive intramolecular stabilization. MQ-C273 is likely compatible with DNA binding. |
| C275 | 3 | 0 | S10/H2 link | MQ-C275 is observed in the DNA-free structures forming stabilizing interactions. This MQ modification is likely incompatible with DNA binding. |
| C277 | 4 | 10 | N-terminus of H2 helix | MQ-C277 is observed in both DNA-free and DNA-bound structures, forming stabilizing interactions. The MQ-mediated effect is highly significant in supporting the p53-DNA interface of the tetrameric complexes. |

[a]Based on the resolved MQ-Cys conjugates shown in Table 1.
[b]N/A stands for "Not Applicable".

DNA without MQ (see Methods). A role of MQ transiently increasing the affinity to DNA in the co-crystallization of R273H-DNA cannot be excluded, although crystals of low-affinity complexes can be obtained due to the close proximity of the two components during crystallization. Soaking of preformed crystals of R273H-DNA, R282W-DNA, R273C/S240R-DNA and of wt-DNA in MQ solutions, led to MQ conjugates at residues C124, C229 and C277.

MQ-C124 conjugates are clearly defined in the p53-DNA-MQ structures but not in the current DNA-free structures, showing only density traces (Table 1). The very low level of this conjugate is likely caused by the restricted accessibility of C124 to MQ in the crystal structures of the free p53 mutants. However, computational studies of the p53 core domain have shown that C124 is located at a transiently accessible L1/S3 pocket, and could be used as a target for reactivating p53 mutants[40]. Such transient conformations could be attained in solution and in cells, but less likely in the crystalline environment. MQ-C229 and MQ-C277 conjugates are also present in the structures of the DNA-free mutants, whereas MQ-C275 observed in the free mutants is not compatible with DNA binding as described above.

MQ-C273 is likely compatible with DNA binding as it does not sterically interfere with the bound DNA (modelled in Supplementary Fig. 3a). This is supported by data from a band-shift assay performed on cell extract, showing significant DNA binding of the oncogenic R273C mutant after treatment with PRIMA-1[23]. The model does not imply that MQ-C273 contributes significantly to complex stability because its shortest distances with DNA are non-bonded VdW contacts (3.5–3.8 Å) between the MQ carbonyl oxygen and the DNA phosphate oxygens. Yet, this does not exclude an alternative conformation of MQ-C273 that might lead to stabilizing interactions with DNA. C273 is not modified in the structure of R273C/S240R-DNA-MQ, because the potential space for MQ bound to C273 is occupied by the second-site mutation R240, forming alternative interaction with the DNA backbone, compensating for the loss of R273-mediated hydrogen bonds[16] (Supplementary Fig. 3b).

It is however surprising that C182, being the cysteine with the largest solvent accessible surface area[42,43] is not bound to MQ in any of the p53-DNA complexes. MQ-C182 conjugates in the structures of the free mutants are present only at sites where C182 residues are attached to an outward L2 conformation, directed towards the solvent, and supported by neighboring molecules (Fig. 1 and Supplementary Fig. 1), whereas the thiol groups of C182 residues in the DNA-bound structures are directed away from the solvent, and thus are less accessible to MQ (e.g.[11,41]).

A very different stabilization of the p53 core domain, by a small molecule, was recently reported[51]. This study demonstrates that arsenic trioxide (ATO) rescues structural mutants, through a cryptic allosteric site, by arsenic (As) binding to a triad of Cys residues: C124, C135, and C141. The rescue mechanism is revealed in two crystal structures of As-bound p53DBD mutants (G245S and R249S) and supported by a wealth of accompanying assays of these and many other cancer-related mutants. It is reasonable to assume that a similar mechanism leads to the rescue of other structural p53 mutants because As binding is site-specific, and its stabilization modes are highly similar among the six independent p53DBD molecules of the two crystal structures[51].

By contrast, due to its chemical properties, MQ binding to cysteines is variable and dynamic, showing a large diversity in the stabilizing modes of equivalent MQ-Cys conjugates. Thus, MQ binding is not specific in the conventional manner. The MQ-bound structures share the following characteristics, suggesting a "combined specificity" of this molecule. First, MQ binds exclusively to surface residues, and not interfering with key p53 regions incorporating thiol groups, like the structural Zn coordination which is critical in supporting the stability of p53 and its complexes with DNA (e.g.[10,11]). Second, MQ selectivity is higher in the DNA-bound state of p53. Only three MQ-modified cysteines (C124, C229 and C277) are observed in the p53-DNA-MQ structures of both mutant and wt p53 which are well-matched with the 3D architecture of the functional tetrameric complex. Therefore, it is likely that this selectivity occurs at the DNA recognition process which involves the binding of a symmetrical

dimer to one DNA half-site (made of two symmetrically-oriented quarter sites), and followed by the binding of a second dimer to the other DNA half-site, forming a p53-DNA tetramer[52].

C124 accessibility to MQ is significantly greater in the DNA-bound than in the DNA-free state, and hence MQ-C124 is likely selected preferentially at the DNA-bound dimers and tetramers. MQ-C229 conjugates are selected in both DNA-free and DNA-bound states, showing similar binding modes that are more supportive in the later. However, MQ-C277 conjugates, selected in both states, show different binding modes resulting from MQ interactions with DNA. By contrast, MQ-C182 and MQ-C275 observed in the DNA-free mutants, are not selected in the DNA complexes, likely because of limited accessibility of C182 to MQ and incompatibility of MQ-C275 with DNA binding, described above.

On the basis of thermal-stability data and cell-based assays, it has been proposed that C277 contributes to MQ-mediated stabilization of the free core domains of wild-type and mutant p53, and that both C124 and C277 are required for MQ-mediated functional restoration of R175H in tumor cells[39]. The potential roles of the two cysteines are in accord with the structural data (see Supplementary Discussion).

The detailed analysis of the MQ-modified structures shows that MQ-Cys conjugates contribute to local stabilization of p53 regions around the modified sites and to DNA binding by hydrogen bonds, CH···O and VdW interactions, which in turn support MQ retention at specific Cys residues. Direct hydrogen bonds and electrostatic CH···O interactions[53–55] appear to play a major role in mutual stabilization by virtue of the distinctive atomic configuration of the bound MQ. In particular, novel hydrogen bonds are formed between the MQ bound to C277 and DNA bases in the complexes of the two mutants, R273H and R282W.

The crystal structures of tetrameric p53 core domains bound to DNA are most relevant to the physiological function of p53 as a transcription factor. These structures are the least affected by crystal packing because the p53 dimers and hence tetramers are stacked continuously along the DNA helices, forming a parallelogram shape similar to that observed in a single-particle cryo-EM structure of full-length p53 bound to DNA[56].

The comparative structural analysis reveals two mechanisms that act concurrently to stabilize the complexes between p53 tetramers and their DNA response elements. MQ-C277 stabilizes the protein-DNA interface, whereas MQ-C124 and MQ-C229 support the interface between p53 dimers (Fig. 6). Thus, although the stabilizing interactions mediated by MQ vary among the tetrameric complexes, they lead to a common global effect in supporting the structural stability of functional p53-DNA complexes. Similar mechanisms might reactivate other cancer-related mutants (see Supplementary Discussion).

To conclude, our structural data demonstrate that three cysteines: C124, C229 and C277 contribute in different ways to MQ-mediated stabilization of mutant and wild-type p53 bound to their DNA response elements in a dynamic process. These data uncover a common platform by which a small molecule, via its active product MQ, is capable of restoring mutant p53 as a tumor suppressor, and hence could serve as a universal drug for treating mutant p53-dependent cancer.

## Methods

**Production and purification of wt-p53DBD.** pET-27b-p53DBD plasmid encoding the core domain of human p53 (residues 94–293, described previously[11]) was transformed into *E. coli* BL21 (DE3) cells. Few colonies were added to 10 ml of lysogeny broth/Luria-Bertani medium (LB medium) supplemented with antibiotics (20 μg/ml kanamycin) and grown for 4 h at 37 °C. This starter culture was added to 1.5 liter of LB media with 20 μg/ml kanamycin. Cells were grown at 37 °C to $OD_{600}$ of 0.5, followed by induction of p53DBD expression by 0.5 mM IPTG at 15 °C for about 18 h. Further purification steps were carried out at 4 °C. Cells were harvested

by centrifugation and the cell pellet (nearly 4 grams) was re-suspended in 50 ml of lysis buffer containing 20 mM sodium citrate, pH 6.1, 100 mM NaCl, 10 μM $ZnCl_2$, 10 mM DTT (buffer A). Cell suspension was sonicated for 5 min (VCX 750, Sonics) using pulse regime (5 sec 'on' and 5 sec 'off') and the soluble fraction was separated by ultracentrifugation. The supernatant was filtered through 0.22 μm pore size filter (EMD Millipore) and half of it was loaded onto a 5 ml HiTrap SP cation exchange column (GE Healthcare) which was pre-equilibrated with 10 column volumes (CV) of buffer A. The protein was eluted with a linear gradient of salt (0.1–1.0 M NaCl) from Buffer A to Buffer B (20 mM sodium citrate, pH 6.1, 1 M NaCl, 10 μM $ZnCl_2$, 10 mM DTT) over a course of 15 CV. The same procedure repeated for the other half of the supernatant. Protein fractions of a total volume of nearly 13 ml containing p53DBD were collected and concentrated to a volume of about 4 ml by 10 kDa MW cutoff filter (Amicon Ultra-15, EMD Millipore) and centrifuged. The supernatant was loaded in batches of 2 ml on a HiLoad 16/60 Superdex 75 PG gel filtration column (GE Healthcare) in buffer containing 20 mM sodium citrate, pH 6.1, 150 mM NaCl, 10 μM $ZnCl_2$, 10 mM DTT. The protein was eluted at retention volume of ~76 ml. Part of the wt-p53DBD protein fractions were submitted to dialysis in order to exchange the 10 mM DTT reducing agent by 2 mM TCEP (Tris (2-carboxyethyl) phosphine) using floating dialysis tubes (GeBA, GeBaFlex-tube, MWCO 6–8 kDa). The collected protein fractions were concentrated to 3–7 mg/ml with 10 kDa MW cutoff filter (Amicon Ultra-15, EMD Millipore), aliquoted and flash-cooled in liquid nitrogen and stored at −80 °C until further use.

**Production and purification of p53DBD mutants.** pET-27b-p53DBD plasmid described above was used as a template in a QuickChange® site-directed mutagenesis reaction (Agilent). The resulting plasmids incorporated the sequences encoding the mutants R273C, or R273H (described by Eldar *et al.*[16]) and R282W. Primer sequences used are as follows (changed codons highlighted):
    5′ GG AAC AGC TTT GAG GTG CAC GTT TGT GCC TGT CCT GGG for R273H,
    5′ G AAC AGC TTT GAG GTG TGC GTT TGT GCC TGT CCT G for R273C, and
    5′ C TGT CCT GGG AGA GAC TGG CGC ACA GAG GAA GAG AAT CTC for R282W.
The plasmid encoding R273C was used as a template for a second mutagenesis reaction to obtain a plasmid encoding R273C/S240R, using the primer 5′ C AAC TAC ATG TGT AAC AGG TCC TGC ATG GGC GGC. The purification protocols used for wt-p53DBD were followed to obtain the R273H and R273C mutants. To prevent aggregation of the R282W mutant after gel filtration, the protein was dialyzed overnight against the same buffer used for the gel filtration step (see above) but without NaCl.

**DNA oligomers and MQ samples used for crystallization.** As in our previous studies[11,16,18,41], two self-complementary DNA oligonucleotides of sequences 5′-CGGGCATGCCCG-3′ or 5′-tGGGCATGCCCGGGCATGCCC-3′, carrying one or two DNA half-sites (consensus sequences underlined, lower case standing for 5′-overhung thymine nucleoside), were purchased (after standard desalting and lyophilization) from IDT (Integrated DNA Technologies, Israel) and purified by ion-exchange chromatography. The DNA oligomers were then dialyzed against water, lyophilized and re-dissolved in water to obtain a concentration of 20 mg/ml.

MQ (2-methylene-quinuclidin-3-one), the reactive electrophile spontaneously formed from APR-246, was provided by Aprea Therapeutics AB (Solna, Sweden). The yellowish solution was diluted into DDW to a stock concentration of 100 mM, aliquoted and stored at −80 °C.

**Crystallization, co-crystallization and soaking experiments.** To avoid potential competition for MQ binding between Cys residues and DTT, preliminary crystallization experiments of MQ bound to wt-DNA complex were performed in buffer containing TCEP (Tris (2-carboxyethyl) phosphine) as the reducing agent, instead of DTT. Although crystals of wt-DNA-MQ diffracting to high resolution were achieved, the crystallization process in the presence of TCEP was less efficient than that with DTT, as only very few conditions yielded suitable crystals which were highly fragile, making their handling during soaking in MQ solutions and/or mounting very difficult and uncertain. It was therefore decided to use DTT in further experiments to obtain MQ-modified crystals. These turned out to be more successful in terms of efficiency and quality. Crystals grown from solutions with either TCEP or DTT showed the same MQ-modified cysteines.

Crystals of mutant and wt proteins and their complexes with DNA were grown at 19 °C by the hanging-drop vapor-diffusion method[57] from 4 μl of drops (including 2 μl protein/DNA solution and 2 μl reservoir solution) equilibrated against 0.5 ml of reservoir solution. The protein concentrations in the 4 μl drops varied from 1.9 to 2.5 mg/ml. Initial crystallization experiments were performed using Hampton Research PEG/Ion™ and PEG/Ion 2™ crystallization screens. Crystallization conditions were optimized using homemade solutions. Crystals appeared after three to four days as in our previous studies. Final crystallization conditions are in Supplementary Table 1.

Co-crystallization and soaking approaches were systematically tried to obtain crystals containing MQ. In the co-crystallization procedure, MQ solutions at different concentrations ranging from 0.5 to 100 mM were added to the protein or

the protein/DNA solutions and incubated on ice for no more than one hour prior to crystallization. At low MQ concentrations, no MQ-related electron density was detected. At high MQ concentrations, crystals grew smaller and diffracted poorly. In the soaking procedure, crystals were transferred into a new host solution mimicking the drop condition but containing also MQ. Different MQ concentrations ranging from 10 to 100 mM, and incubation times from 5 min to two days were tried. Typically, high MQ concentrations degraded the integrity of the crystals and their diffraction. In most cases, long incubation times had similar effects. In order to stabilize the crystals prior to soaking, few crystals were cross-linked using glutaraldehyde (GA)[58,59]. This was done (under a chemical hood) by placing 2.5 μl drops made of 25% GA on the cover slips in the vicinity of the hanging drops. The cover slips were re-sealed, and the drops were equilibrated for 10 min. Then, GA was inactivated by adding 1 M NaOH, and the crystals were transferred to MQ soaking solutions. This method was successful for one crystal structure, R273H-MQ (I) (Table 1, Supplementary Table 1).

**Data collection and processing**. Before data collection, crystals were transferred into a cryo-protectant for few minutes. Paratone-N oil (Hampton Research) was successfully used for most of the crystals. When needed, alternative cryo-protectant solutions were prepared mimicking the stabilizing mother liquor supplemented with glycerol or ethylene glycol at concentrations leading to efficient cryo-protection. The cryo-protected crystals were mounted on Hampton Research CryoCapHT nylon loops and flash-cooled in a stream of nitrogen gas at 100 K using an Oxford Cryostream 700-series for in-house data collection or plunged into liquid nitrogen and stored for further measurements.

Crystals were checked at the Weizmann Institute X-ray Crystallography Laboratory on a Rigaku R-AXIS IV + + Imaging Plate detector mounted on Rigaku MX-007HF rotating-anode generator, with CuKα radiation (1.54178 Å) focused by a VariMax optics HF system. Diffraction data were collected at the European Synchrotron Radiation Facility (ESRF, Grenoble, FRANCE) at beamlines ID23-1, ID23-2, ID29 and ID30A-3 or at our home facility. At the synchrotron facility, MxCuBE[60] was used for data collection, and systematically optimized by the characterization and strategy program EDNA[61], taking into account radiation damage. Complete data set from a single crystal was collected in each case. The data were indexed, integrated and scaled with HKL-2000[62]. Data statistics is in Supplementary Tables 2–5.

**Structure determination and refinement**. The atomic models of enantiomers formed by MQ binding to amino acids (referred as MQr and MQs), were built and submitted to energy minimization using Chimera[63], to optimize bond lengths and angles. For each of the two enantiomers, a cif file was produced with eLBOW from Phenix[64] and used during the refinement.

Prior to structure determination, each data set was subjected to lattice and twinning analysis using Xtriage from the Phenix package[65]. The new crystal structures were isomorphic to previously published structures of the p53 core domain or its complex with DNA[11,16,41]. Two of the four crystals of the modified free mutants, R273H-MQ (II) and R273C-MQ II), all belonging to the monoclinic space group P2₁ with β≈90°, were identified as pseudo-merohedral twins observed previously[16].

To minimize bias and errors, the new structures were solved independently by molecular replacement using Phaser[66] implemented in the Phenix package[65]. For the structures of the MQ-modified p53 mutants, the crystal structures of the corresponding mutant proteins (PDB codes 4IBS or 4IBQ, chain A), were used as search models. For the structures of the new protein-DNA and MQ-modified protein-DNA complexes, the crystal structure of the human p53 core domain (PDB code 2AC0, chain A) was used as a search model. In addition, each refinement was initiated with a slow-cooling simulated annealing cycle using Phenix[65]. Successive rounds of model building, manual corrections with Coot[67] and refinement with Phenix[65] or with Refmac5[68] (the later for the twinned structures) were performed to build the complete models. Automatically assigned TLS groups were systematically tested and used in cases of clear improvement of the structure refinement. When the quality of the crystallographic data permitted, individual anisotropic ADPs were used. For each crystal structure, a randomly created R-free data set was used and kept throughout the process to monitor the refinement progress. For the protein-DNA complexes, the DNA molecules were traced into SigmaA and SigmaD electron density maps using Coot[67]. The solvent molecules were initially added automatically, then checked and adjusted using Coot[67] after each cycle of refinement.

Bulks of electron density (based on SigmaA and SigmaD maps) compatible with bound MQ were systematically searched within the asymmetric unit using Coot ad-hoc option[67]. All but one of these bulks of density were located at the vicinity of cysteines (at a distance around 1.8 Å) and no free MQ was found in the solvent. Covalently-bound MQ molecules were docked into the electron density, and refined together with the other components of the structure, including MQ occupancies and temperature factors. To achieve an accurate description of these parameters, MQ-free residues were simultaneously refined with the MQ-bound ones. The identity of the enantiomer was checked throughout the refinement. In some cases, alternative MQ enantiomers and/or alternative orientations were modeled to account for the observed electron density.

During the refinement, only MQ molecules exhibiting good fitting to the electron density and reasonable temperature factors and occupancies were kept in the modeled structure. The successfully modeled MQ sites following this protocol are in Table 1. All correspond to MQ covalently bound to Cys residues except for a single MQ bound to K101 observed in monomer A of wt-DNA-MQ (P1). In addition to the well-defined sites, Table 1 lists sites likely containing MQ (based on the corresponding electron-density bulks), but not included in the refined structures because of failing to meet our acceptance criteria. Several examples of MQ-Cys conjugates within their electron density maps are in Supplementary Fig. 9. MolProbity[69] was used to check and analyze the results of each refinement cycle as well as the final crystal structure prior to deposition. Refinement statistics is in Supplementary Tables 6–9.

All the figures presenting structural information were done by PyMOL[70].

For simplicity, the names of the MQ-modified structures are referred by the protein name or its DNA complex and followed by MQ (e.g. R273H-MQ, R273H-DNA-MQ). The modified residues within each structure are referred as MQ-residue (e.g. MQ-Cys, MQ-C277). The detailed nomenclature of the three types of modified amino acids in the current structures (MQr-Cys, MQs-Cys and MQr-Lys) and their chemical information are in Supplementary Note 1.

**Reporting Summary**. Further information on research design is available in the Nature Research Reporting Summary linked to this article.

## Data availability

Crystal structures data have been deposited in the Protein Data Bank under the accession codes: 7B47 for R273H-MQ (I), 7B48 for R273H-MQ (II), 7B49 for R273H-DNA-MQ, 7B4A for R273H-DNA, 7B4B for R273C-MQ (I), 7B4C for R273C-MQ (II), 7B4D for R273C/S240R-DNA-MQ, 7B4E for R282W-DNA-MQ, 7B4F for R282W-DNA (I), 7B4G for R282W-DNA (II), 6ZNC for wt-DNA-MQ (I), 7B4N for wt-DNA-MQ (II), 7B4H for wt-DNA-MQ (III), 7B46 for wt-DNA-MQ (P1). Previously published crystal structures used in this study are available from the Protein Data Bank under the accession codes: 2AC0, 4IBQ, 4IBS, 4IBV, 3IGL, 5MCV, 5MCW. Data supporting the findings of this manuscript are available from the corresponding authors upon reasonable request.

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

## Acknowledgements

We thank Professor K. Wiman for motivating us to pursue this long-term research. We thank our colleagues M. Eisenstein for help in modeling of MQ conjugates and A.

Kapitkovsky for help in protein production and purification. We thank S. Semenov for his guidance regarding the nomenclature of organic molecules and chemical modifications of amino acids. We also thank the staff at the ESRF (Grenoble, France) for their assistance, in particular D. Flot, M. Nanao and D. de Sanctis. This work was supported by funding from Aprea Therapeutics AB, and the Kimmelman Center for Biomolecular Structure and Assembly (Weizmann Institute of Science) to Z.S.

## Author contributions

O.D., L.A., H.R. and Z.S. participated in planning the research. O.D., D.G. and Y.D. produced and purified the proteins. H.R. directed the X-ray crystallographic work. O.D., D.G., Y.D., H.R. and Z.S. participated in the X-ray crystallographic work and analyzed the crystal structures. O.D., D.G., L.A., H.R. and Z.S. contributed to manuscript preparation.

## Competing interests

L.A. is an Employee and equity holder of Aprea Therapeutics, developing eprenetapopt. All other authors declare no competing interests.
