## [Peer Review File · Nature Communications]

Structural basis of reactivation of oncogenic p53 mutants by a small molecule: methylene quinuclidinone (MQ)REVIEWER COMMENTS

Reviewer #1 (Remarks to the Author):

APR-246 is in clinical trials in blood cancers as an agent to rescue the tumor suppressor function of mutant p53. This study uses X-ray crystallography to assess how covalent binding of the active product MQ may exerts its effects. Overall, the work is a tour de force in crystallography with structures of R273H, R273C, R282W, R273C/S240R and WT p53 DNA-binding domain, of which some are solved with and without bound DNA. I congratulate the authors on their achievements here. Similar structural work on p53 rescue by the covalent binding of arsenic trioxide was recently published in *Cancer Cell* 39, 225-239 (Feb2021). That study contained a wealth of accompanying assays for mechanistic insight that is missing in this submission on MQ. Instead, the authors conducted only crystallographic work, which is disappointing. As a result, the aims, discussion and take home conclusions from the current work remain somewhat vague. This may reflect reality – in that the mechanism of MQ1 is multifactorial and complex(?). It is apparent from the number of modification sites spanning Cys and Lys residues that MQ is extremely promiscuous. Previous literature have assigned at least part of its mechanism of action to be via glutathione biology rather than p53. Indeed, new work in BioRxiv suggests the cystine/glutamate transporter SLC7A11 as the superior determinant of APR-246 response.

Major comments

1) Being somewhat generous, the lack of accompanying assay data may reflect that these data already exist in the literature? However, if this is the case, the authors fail to draw on this resource to frame their study. The authors give only 2-3 lines introduction on past mechanistic understanding of APR-246/MQ, mentioning induction of p53-target genes, as well as apoptosis in p53 mutant and WT backgrounds. More introduction description would be helpful e.g. learnings on thermal stability, which cysteines are previously reported to be modified, which p53 mutants have or have not been studied with APR-246 before compared to their work? What are the learnings from clinical trials? These points lead into comment 2 below.

2) Definition of the aims of the study (introduction line 80) – this is only defined as “To uncover the structural basis of rescuing p53 mutants by MQ”. Can the aim(s) be defined any more specifically based on past learnings, from point 1 above? This would then provide a focus and line of investigation for the conclusions/discussion. For example, Zhang et al. previously claimed that “APR-246 reactivates mutant p53 by targeting cysteines 124 and 277”. This work is cited in the discussion (rather than the introduction). Does this structure-based study assess their claim? (in the discussion, it is not clear what the authors conclude e.g. were Zhang et al correct, partially correct or wrong – see point 3 below).

3) The conclusions of the study are described at length, but still left me feeling they were vague and incomplete. For example, line 453 “MQ bound to C273 is likely compatible with DNA binding (modelled in Supplementary Fig. 3), but in the absence of experimental data, it is not clear whether this conjugate contributes to the stability of the complex.” I would expect the authors to address this question with an assay. If they lack capability, one assumes a collaborator would manage. The remainder of the conclusions are somewhat general. The general take home messages were (i) that some cys-MQ1 adducts allow DNA contact and (ii) some Cys-MQ adducts allow intramolecular contacts between p53 subunits within the DNA-bound p53-tetramer (and iii not addressed here is thermostability). It is not 100% clear, but I suspect the authors believe it is a combination of MQ1 mechanisms that is important, and therefore activity cannot be assigned to one cysteine, or one interaction? Does the study suggest any specific p53 cancer mutants that may be more amenable to rescue by MQ than others? Perhaps they could also add a summary table with each Cys listed and the conclusions for that residue (relative importance and type of effect mediated?).

4) The abstract concludes “The current findings provide a structural framework for the design of new compounds targeting specific sites of mutant p53 for its reactivation in cancer therapy.” Yet, this topic is not discussed in the manuscript. To justify this as a primary abstract point, I would expect a few

comments on it in the discussion? Do the structures show potential aspects for scaffold improvement, or is MQ perfect, or are there aspects of negative design to improve selectivity?

5) Can the authors include in their discussion some comments on MQ selectivity and potential physiologically relevant sites. The study shows a large diversity in p53 Cys sites targeted and MQ appears high unselective. Some sites appear beneficial from the structures, but some are detrimental e.g. Line 174 states MQ-C275 conjugates are not compatible with DNA binding. How do the authors see the likely balance or significance of this C275 binding with other positive effects? Crystal soaking and co-crystallisation use huge concentrations of drug and protein that would not be expected in cells. Therefore, how relevant are the patterns of covalent binding in crystallography – this question relates to either performing their own assays, or drawing on literature (see also point 1)?

Minor comments

1) The authors don't comment on the choice of mutants studied. I assume they tried a much wider panel of p53 mutants, but others failed to crystallize. It would be valuable for the authors to add one sentence on this (negative data is still informative).

2) Numbering error. Lines 267-269 describe MQ-C124 orientations in the R282W-DNA-MQ structure referring to Supplementary Fig. 4a-d. However, the R282W structures only appear in Supplementary Fig. 4f.

3) The figures use red dashed lines for H bonds and blue dashed lines for CH-O van der Waals interactions. My preference would be for the blue lines to be removed. I am not convinced they are significant. However, I leave this to the editors, other reviewers and authors.

4) I can understand that the authors had challenges in finding the right balance between figures chosen for the main manuscript and those for supplemental. All the opening figures only show a handful of residues giving the reader little sense of their relative position in the complete p53 domain. It would be helpful to have a Figure 1a panel resembling Suppl Fig 2A but showing the relative positions of all the cys positions + mutants R282, R273 etc covered in the study. Conversely, most main figures simply show the same data across multiple subunits in a crystal asymmetric unit. E.g. Figure panels 1a-d are essentially all nearly identical. Moreover, the summary text for these panels (line 145) states "the role of MQ-C182 on p53 stabilization and hence reactivation is not clear". It is not obvious we need a full picture with 4 duplicate panels just for this point.

5) Line 264 "R282-DNA-MQ". Shouldn't this be R282W?

6) Line 458 "It is however surprising that C182, being the cysteine with the largest solvent accessible surface area in any of the p53-DNA complexes." This sentence is missing a middle part. Surprising that "what?". It appears to relate to the last sentences of the previous paragraph, but has lost context.

7) Can the 10 mM DTT reduce the MQ bonds? Does this explain the failure to see MQ in the WT and R273H p53 soaked crystals?

Reviewer #2 (Remarks to the Author):

This manuscript describes highly detailed structural studies of the binding modes of methylene quinuclidinone, the biologically active product of the anticancer drug candidates PRIMA1 and APR246, to wild-type and cancer-driving mutant p53 proteins. The crystal structures provide new insights about features that may give rise to the reactivation of impaired p53 mutants. APR246 is a promising drug candidate, and the information provided may guide further compound optimization in the future. The manuscript is suitable for publication with two minor modifications.

(1) Methylene quinuclidinone acts by Michael addition, and as such, is a promiscuous compound

lacking specificity. The general readers would benefit from a short discussion of this issues and if possible suggestions of how better specificity may be achieved and toxic effects mitigated.
(2) Lines 458-459: The sentence was probably truncated unintentionally. Please correct.

Response to Reviewers

We thank the reviewers for their insightful and constructive criticism. It has guided us in revising and improving the manuscript as described below (cited references are in alphabetical order below). Also included here, at the end of the **Response**, is the **Supplementary Discussion** from the **Supplementary Information**. The revised parts, based on the reviewers' comments, in the Main Text and the Methods are highlighted in red.

Reviewer #1 (Remarks to the Author):

APR-246 is in clinical trials in blood cancers as an agent to rescue the tumor suppressor function of mutant p53. This study uses X-ray crystallography to assess how covalent binding of the active product MQ may exert its effects. Overall, the work is a tour de force in crystallography with structures of R273H, R273C, R282W, R273C/S240R and WT p53 DNA-binding domain, of which some are solved with and without bound DNA. I congratulate the authors on their achievements here. Similar structural work on p53 rescue by the covalent binding of arsenic trioxide was recently published in *Cancer Cell* 39, 225-239 (Feb2021). That study contained a wealth of accompanying assays for mechanistic insight that is missing in this submission on MQ. Instead, the authors conducted only crystallographic work, which is disappointing. As a result, the aims, discussion and take home conclusions from the current work remain somewhat vague. This may reflect reality – in that the mechanism of MQ1 is multifactorial and complex (?). It is apparent from the number of modification sites spanning Cys and Lys residues that MQ is extremely promiscuous. Previous literature have assigned at least part of its mechanism of action to be via glutathione biology rather than p53. Indeed, new work in BioRxiv suggests the cystine/glutamate transporter SLC7A11 as the superior determinant of APR-246 response.

Response

The determination of 14 new crystal structures presented in the manuscript has been indeed “a tour de force undertaking in crystallography”, with respect to crystallization experiments to achieve crystals diffracting to high resolution, and their analysis to obtain an accurate determination of the occupancy levels of the MQ-Cys adducts, as well as their interaction modes. The binding of MQ to p53 appears to be highly driven by chemical reactivity and is reversible, therefore we attempted to achieve structural data on the same p53 molecules under distinctly different conditions (e.g. soaking crystals into MQ solutions, co-crystallization, and large ranges of MQ concentration and incubation times). The structural data is very extensive and perhaps could be published in more than one paper, but we decided to include all the structural data in one manuscript in order to present a comprehensive and integrated atomic-resolution view on the potential effects of a drug molecule that is currently undergoing clinical trials.

The binding of MQ to surface p53 cysteines is variable and dynamic in contrast to the highly site-specific, simultaneous and strong binding of arsenic to three Cys residues

in p53, stabilizing structural p53 mutants (Chen *et al.*, 2021). The rescue mechanism of arsenic trioxide (ATO) via a cryptic binding site in p53 is revealed in two crystal structures of arsenic-modified p53DBD mutants (G245S and R249S), and supported by a wealth of assays of these and other mutants. In this case, because arsenic binding is site-specific and the stabilization modes are highly similar among the six independent p53DBD molecules as shown by the two crystal structures, it is reasonable to assume that a similar binding mechanism led to the rescue of other structural p53 mutants, shown to be stabilized by ATO.

In contrast to the site-specific and essentially irreversible binding of arsenic, MQ binds reversibly, and a large diversity is observed in its stabilizing modes of the same cysteines in different p53 proteins. The interactions are “multifactorial and complex”, and hence requires a comparative analysis of a wealth of structural data for putting forward a common reactivation mechanism of non-functional p53 proteins. Another consequence of the reversible binding of MQ is that a much higher excess of this molecule is required for stabilization compared to ATO, although each entity stabilizes the corresponding p53 core-domain by binding to its specific native state. The rapid reactivity of MQ with cysteines, although reversible, also explains why a high cellular level of glutathione is a resistance marker, and why high expression of SLC7A11 comes up as a marker for resistance (referred to in the **Introduction**).

The aim of the current work was to use crystallography and structural analysis in an attempt to find explanations for the observed stabilization of both wild-type and p53 mutants. We were very fortunate to obtain the crystal structures of functional DNA complexes of the core domains of wild-type p53 and of DNA-contact and structural mutants, as these three types of variants have previously been shown to be rescued by MQ. Clearly, crystallography is not the method to identify mutants that are not rescued (which do exist), nor to investigate the strength of MQ-mediated stabilization. Thus, we aim to present a structural explanation without attempting to assess the magnitude of the stabilization of different mutants – suffice to say that quantitative differences in stabilization are to be expected among the various reported proteins. We therefore think that biochemical analyses are beyond the scope of this manuscript and believe that our findings will inspire researchers to examine our proposed reactivation mechanism by a wealth of functional assays in a large number of cancer-related p53 proteins.

Major comments

1) Being somewhat generous, the lack of accompanying assay data may reflect that these data already exist in the literature? However, if this is the case, the authors fail to draw on this resource to frame their study. The authors give only 2-3 lines introduction on past mechanistic understanding of APR-246/MQ, mentioning induction of p53-target genes, as well as apoptosis in p53 mutant and WT backgrounds. More introduction description would be helpful e.g. learnings on thermal stability, which cysteines are previously reported to be modified, which p53 mutants have or have not been studied with APR-246 before compared to their work?

What are the learnings from clinical trials? These points lead into comment 2 below.

Response to comment 1.

There are different types of data on more than 100 different p53 mutants treated by APR-246. A major part of these data is related to different efficacy assays in cells. The starting point for the present study is that there are effects of APR-246 on most of the mutants, but to different degrees. As suggested, we have added to the **Introduction** the results of previous assays on APR-246, PRIMA-1 (the two alternative prodrugs of MQ) and MQ that are relevant to the current structural findings, and in more detail to the **Supplementary Discussion** (described also in response to comments 2 and 3). Relevant data has been included from Bykov *et al.* (2002), Wassman *et al.* (2013), Zhang *et al.* (2018), and Demir *et al.* (2020). We also refer to TP53 mutant data in relation to clinical effect studied in two clinical trials: Lehmann *et al.* (2012) and Sallman *et al.* (2021).

2) Definition of the aims of the study (introduction line 80) – this is only defined as “To uncover the structural basis of rescuing p53 mutants by MQ”. Can the aim(s) be defined any more specifically based on past learnings, from point 1 above? This would then provide a focus and line of investigation for the conclusions/discussion. For example, Zhang *et al.* previously claimed that “APR-246 reactivates mutant p53 by targeting cysteines 124 and 277”. This work is cited in the discussion (rather than the introduction). Does this structure-based study assess their claim? (in the discussion, it is not clear what the authors conclude e.g. were Zhang *et al.* correct, partially correct or wrong – see point 3 below).

Response to comment 2.

The aim and driving force of our study are directly related to the far-sighted statement made by Lambert *et al.* (Cancer Cell, 2009) after showing that PRIMA-1 is converted to MQ that forms adducts with thiols in mutant p53, and that covalent modification of cysteines in mutant p53 induces apoptosis in tumor cells. Here is the quotation from that article: “Analysis of PRIMA-1-modified mutant p53 by X-ray crystallography should ultimately provide information about the structural consequences of cysteine alkylation”. We’d like to point out that our structural studies started well before the published studies of Zhang *et al.* (2018), and hence did not rely on this, or attempted to assess their data. But as mentioned above, a summary of their results was added to the **Introduction**, and described in more detail in the **Supplementary Discussion** (below).

3) The conclusions of the study are described at length, but still left me feeling they were vague and incomplete. For example, line 453 “MQ bound to C273 is likely compatible with DNA binding (modelled in **Supplementary Fig. 3**), but in the absence of experimental data, it is not clear whether this conjugate contributes to the

stability of the complex.” I would expect the authors to address this question with an assay. If they lack capability, one assumes a collaborator would manage.

The remainder of the conclusions are somewhat general. The general take home messages were (i) that some cys-MQ1 adducts allow DNA contact and (ii) some Cys-MQ adducts allow intramolecular contacts between p53 subunits within the DNA-bound p53-tetramer (and iii not addressed here is thermostability). It is not 100% clear, but I suspect the authors believe it is a combination of MQ1 mechanisms that is important, and therefore activity cannot be assigned to one cysteine, or one interaction?

Does the study suggest any specific p53 cancer mutants that may be more amenable to rescue by MQ than others? Perhaps they could also add a summary table with each Cys listed and the conclusions for that residue (relative importance and type of effect mediated?).

Response to comment 3.

The discussion and conclusions of our study were described at length because the structural data is enormous. In particular, we think that it is important to understand the following. (1) Why the modified Cys residues appear to vary between crystals of the same p53 system obtained under different crystallization procedures (e.g. soaking crystals in MQ, or by co-crystallization with MQ). (2) Why are there differences in the modified Cys residues between the DNA-free and the DNA-bound structures. Therefore, such topics are presented at length in the discussion. These issues are related directly to the factors that determine *selectivity* of MQ binding at specific sites described in response to comment 5.

MQ bound to C273 and its compatibility with DNA binding. Whereas the crystal structures provide detailed information on potential MQ interactions, crystallization is still unpredictable. Attempts to obtain crystals of R273C-DNA-MQ were unsuccessful. The DNA-free structures of R273C-MQ show high occupancy levels and similar binding modes in all eight p53 monomers (**Table 1, Fig. 5**). Hence, the structure of a single R273C-MQ monomer was compared with two of our previously reported wt-DNA structures, suggesting that MQ-C273 is compatible with DNA binding, in the sense that it does not interfere with DNA binding (unlike that of MQ-C275). The model does not suggest that it contributes to complex stability because the distances in this model are non-bonded van der Waals contacts (3.5-3.8 Å) between the carbonyl oxygen of MQ and the phosphate oxygen atoms of the DNA backbone. Yet, alternative orientation of the bound MQ relative to DNA might lead to favorable interactions between MQ-C273 and DNA. This issue is clarified in the main **Discussion**. However, either option (being only compatible with DNA binding or directly contributing to DNA binding) cannot be assessed in the absence of structural data. Unfortunately, a DNA binding assay will not provide the required information regarding the potential effect of MQ-C273 on the corresponding R273C-DNA complex, because in such an assay the specific MQ-modified Cys residues are not known. However, a band-shift assay performed on cell extract incorporating R273C showed significant DNA binding of this mutant upon treatment by PRIMA-1 (Bykov *et al.*, 2002, supplementary Table B). According to our structural reactivation

mechanism, restoration of DNA binding to R273C mutant is likely achieved by the three natural Cys residues (at positions 124, 229 and 277) bound to MQ (described also in the **Supplementary Discussion** below).

The proposed reactivation mechanism. The proposed reactivation mechanism based on the p53-DNA-MQ structures is indeed a combination of two MQ-mediated mechanisms, and therefore activity cannot be assigned to one cysteine, or one interaction. The thermal stability of the modified free monomers and the complexes with DNA is likely affected by favorable interactions, mediated via several MQ-bound cysteines, shown in the corresponding crystal structures. These points and their relation to the published assays are clarified in the main **Discussion** and in the **Supplementary Discussion**.

Specific p53 cancer mutants that may be more amenable to rescue by MQ than others. Based on the available structural data (the current MQ-bound structures, previously published structures of wt and p53 mutants) and the published assays, we propose the potential effects of MQ on other p53 cancer mutants (described in the **Supplementary Discussion**).

Summary Table with each Cys listed and the conclusions for that residue. As suggested, we have added a summary **Table 1C** to the Main Text, including the number of the MQ-modified Cys residues in each specific group, their locations in the core domain and short summaries on their effects and significance.

It should be emphasized that the single MQ bound to Lysine (K101), observed in the current study, was captured in one monomer of the wt-DNA-MQ (P1) structure as a result of stabilizing interactions between two neighboring wt-DNA tetramers in the crystal. Hence, MQ-K101 is unlikely to occur under physiological conditions. This issue is clarified in the Main Text in relation to the inter-dimer interface involving MQ-C229, and in the legends of **Fig. 8** and **Supplementary Fig. 7**.

4) The abstract concludes “The current findings provide a structural framework for the design of new compounds targeting specific sites of mutant p53 for its reactivation in cancer therapy.” Yet, this topic is not discussed in the manuscript. To justify this as a primary abstract point, I would expect a few comments on it in the discussion? Do the structures show potential aspects for scaffold improvement, or is MQ perfect, or are there aspects of negative design to improve selectivity?

Response to comment 4.

The sentence on the *design of new compounds* has been removed from the abstract because this is truly a challenging goal. For achieving improved selectivity by derivatives of APR-246/MQ or by new compounds designed to target specific Cys residues, a comprehensive understanding of the factors that contribute to the Michael addition reaction is required. These include “the kinetics of thiol addition reactions, bioactivities, as well as steric and electronic factors that influence the electrophilicity and reversibility of Michael acceptors” (quoted from Jackson P. A. *et al.*, *J. Med. Chem.* 2017, 60, 839-885). In addition, the stability and reversibility of the modified

Cys residue at each specific site depends on its potential interactions with the protein and the DNA response element. Hence, the design of compounds for improved selectivity requires a combined expertise in the fields of medicinal and physical chemistry, structural biology and computational biology (this issue is also discussed in response to Reviewer#2).

5) Can the authors include in their discussion some comments on MQ selectivity and potential physiologically relevant sites. The study shows a large diversity in p53 Cys sites targeted and MQ appears high unselective. Some sites appear beneficial from the structures, but some are detrimental e.g. Line 174 states MQ-C275 conjugates are not compatible with DNA binding. How do the authors see the likely balance or significance of this C275 binding with other positive effects? Crystal soaking and co-crystallisation use huge concentrations of drug and protein that would not be expected in cells. Therefore, how relevant are the patterns of covalent binding in crystallography – this question relates to either performing their own assays, or drawing on literature (see also point 1)?

Response to comment 5.

MQ selectivity and potential physiologically relevant sites. The structural data show the potential sites of MQ-Cys adducts in the free and DNA-bound p53 proteins. These reactions are not specific in the conservative manner where a unique binding mode is presented by the same Cys residues in different proteins (wt and mutant p53). However, the combined effect generated by an ensemble of MQ-modified surface-exposed Cys residues leads to the stabilization of several regions in the free proteins and their complexes with DNA.

The MQ-bound Cys residues (at positions 124, 229 and 277) observed in the various complexes contribute to the integrity of the p53-DNA tetramer by protein-DNA and protein-protein interactions. These are the most physiologically-relevant sites in rescuing mutant p53, leading to common “specific” effects on the stability of p53-DNA tetramers, required to restore p53 function as a tumor suppressor. MQ-C275 is not compatible with DNA binding and therefore not selected in the DNA recognition process. MQ selectivity is discussed in detail in the main **Discussion**. The potential roles of C124 and C277 in the reactivation of mutant p53 based on the current structures are in accord with the corresponding data of Wassman *et al.* (2013), and of Zhang *et al.* (2018), described in the Main Text and in detail in the **Supplementary Discussion**.

The relevance of the patterns of covalent binding in crystallography. X-ray crystallography is certainly not the method to assess physiological concentrations. To obtain quality-diffracting crystals, we used a large concentration range of the various components. To crystallize p53DBD from different systems, we usually use a concentration range of 1-8 mg/ml of protein. MQ binding to thiol groups is highly reversible. Hence, to increase the odds of obtaining a significant fraction (occupancy) of MQ-bound thiols in the crystals, we used a large range of concentration (0.5-100 mM), including high excess of MQ, as well as a large range of incubation time (from

5 minutes to 2 days). For crystallization of p53-DNA complexes, we usually use a large molar excess of DNA over the protein, thereby shifting the equilibrium to the crystallization of the DNA-bound proteins relative to crystallization of the free proteins (e.g. Kitayner *et al.*, 2006; Kitayner *et al.*, 2010). The analysis of the crystals shows that the observed MQ-Cys adducts are dependent on the procedure used (co-crystallization versus soaking) rather than on MQ concentration or incubation time. The effects of MQ concentration and incubation on crystal formation are further detailed in the **Methods**.

Minor comments

1) The authors don't comment on the choice of mutants studied. I assume they tried a much wider panel of p53 mutants, but others failed to crystallize. It would be valuable for the authors to add one sentence on this (negative data is still informative).

Response: We also tried to obtain crystals of the core domain of R175H mutant, but without success because of protein aggregation. However, this failure should not discourage other researchers to keep on trying.

2) Numbering error. Lines 267-269 describe MQ-C124 orientations in the R282W-DNA-MQ structure referring to Supplementary Fig. 4a-d. However, the R282W structures only appear in Supplementary Fig. 4f.

Response: Thanks, the numbering has been corrected in the revised text. It is now **Supplementary Fig. 5a-f**.

3) The figures use red dashed lines for H bonds and blue dashed lines for CH-O van der Waals interactions. My preference would be for the blue lines to be removed. I am not convinced they are significant. However, I leave this to the editors, other reviewers and authors.

Response: CH-O and CH-N contacts are electrostatic interactions contributing to the stabilization of ligand-protein and ligand-nucleic acids structures: e.g. Panigrahi and Desiraju (2007), Desiraju (2011) and Itoh *et al.* (2019). In particular, such interactions play an important role in MQ-mediated stabilization of p53 and its complexes with DNA.

4) I can understand that the authors had challenges in finding the right balance between figures chosen for the main manuscript and those for supplemental. All the opening figures only show a handful of residues giving the reader little sense of their relative position in the complete p53 domain. It would be helpful to have a Figure 1a panel resembling Suppl Fig 2A but showing the relative positions of all the cys positions + mutants R282, R273 etc covered in the study. Conversely, most main figures simply show the same data across multiple subunits in a crystal asymmetric unit. E.g. Figure panels 1a-d are essentially all nearly identical. Moreover, the summary text for these panels (line 145) states "the role of MQ-C182 on p53

stabilization and hence reactivation is not clear”. It is not obvious we need a full picture with 4 duplicate panels just for this point.

Response:

The choice of figures. Indeed, it was challenging to find the right balance between figures chosen for the main text and for the Supplementary Information. To demonstrate the large diversity in the binding modes of the same MQ-Cys in different structures, as well as specific common features, it is important to show the detailed environment and the stabilizing intra- and intermolecular interactions. We tried not to present nearly identical patterns unless observed in crystals of the same protein obtained under different conditions (e.g. soaking versus co-crystallization) exemplified by **Fig. 1 a, b** based on R273H-MQ (I, II) structures, and **Fig. 1 c, d** based on R273C-MQ (I, II) structures.

Figures showing the relative position of the modified cysteines in the complete p53 domain. The relative positions of the MQ-modified cysteines of the monomers from R273H-MQ and R273C-MQ structures are displayed in **Supplementary Fig. 2a, b**. These monomers are compared to the structure of wt p53 monomer bound to DNA in order to estimate their compatibility with DNA binding. The corresponding MQ-Cys adducts in the modified p53-DNA tetramers are shown in **Fig. 6**. However, in view of the small size of the four identical symmetry-related MQ-modified monomers of each complex, we added larger figures of the MQ-Cys adducts of each complex, showing their relative positions in the corresponding core domains (**new Supplementary Fig. 4**).

5) Line 264 “R282-DNA-MQ”. Shouldn’t this be R282W?

Response: Thanks, this has been corrected in the text.

6) Line 458 “It is however surprising that C182, being the cysteine with the largest solvent accessible surface area in any of the p53-DNA complexes.” This sentence is missing a middle part. Surprising that “what?”. It appears to relate to the last sentences of the previous paragraph, but has lost context.

Response: Thanks, this sentence was truncated unintentionally. The corrected new sentence is: “It is however surprising that C182, being the cysteine with the largest solvent accessible surface area, is not bound to MQ in any of the p53-DNA complexes”.

7) Can the 10 mM DTT reduce the MQ bonds? Does this explain the failure to see MQ in the WT and R273H p53 soaked crystals?

Response:

DTT is stable at 2 to 8°C for one week. MQ binding to DTT is reversible as it is to any other thiol group. All the crystallization experiments were done at room temperature (19°C), and MQ-modified crystals appeared after three to four days, so the effect of DTT on the crystals, if any, was negligible.

MQ was not observed in co-crystallization experiments of wt or mutant p53 in the presence of both DNA and MQ, using either DTT or TCEP (Tris (2-carboxyethyl) phosphine) as the reducing agent. Efficient MQ modification was obtained by soaking pre-formed crystals of the various p53-DNA complexes in MQ solutions containing either DTT or TCEP, showing the same MQ-Cys adducts. The quality and handling of the crystals was significantly better by using DTT. This issue is clarified in the **Methods**.

Reviewer #2 (Remarks to the Author):

This manuscript describes highly detailed structural studies of the binding modes of methylene quinuclidinone, the biologically active product of the anticancer drug candidates PRIMA1 and APR246, to wild-type and cancer-driving mutant p53 proteins. The crystal structures provide new insights about features that may give rise to the reactivation of impaired p53 mutants. APR246 is a promising drug candidate, and the information provided may guide further compound optimization in the future. The manuscript is suitable for publication with two minor modifications.

(1) Methylene quinuclidinone acts by Michael addition, and as such, is a promiscuous compound lacking specificity. The general readers would benefit from a short discussion of this issues and if possible suggestions of how better specificity may be achieved and toxic effects mitigated.

Response to comment 1.

About MQ binding specificity: MQ is regarded as a non-specific (promiscuous) compound. However, the various MQ-Cys adducts observed in the current structures, show the following characteristics, granting a certain degree of specificity to this molecule: (1) MQ binds exclusively to surface residues. (2) Only three cysteines (C124, C229 and C277) are modified in the p53-DNA structures of both wt and mutant p53, irrespective of MQ concentration or incubation times. Therefore, it is likely that this selectivity occurs at the DNA recognition process. MQ binding to C277 leads to MQ-mediated interactions between DNA and p53 dimers and tetramers, whereas MQ bound to C124 and C229 support the inter-dimer interface. The specificity issue is discussed in detail in the main **Discussion**.

About achieving better specificity. We have removed the sentence on the *design of new compounds* from the abstract, because this is truly a challenging quest. Using the present structural data to identify modifications of MQ that improve the binding to each of the three cysteines, found to contributing to p53 reactivation, is only a challenging start as the reactivity of the Michael acceptor might be compromised when the molecule is modified. In fact, clinical data indicate a benign toxicity of APR-246, dramatically different from hard alkylators which might depend on the reversible binding kinetics, e.g. Lehman *et al.* (2012) and Sallman *et al.* (2021). Moreover, an improved molecule must have favorable pharmacokinetic and intrinsic drug-like properties (this issue is also discussed in response to Reviewer#1).

⌚

(2) Lines 458-459: The sentence was probably truncated unintentionally. Please correct.

Response: Thanks, indeed truncated unintentionally. The corrected new sentence is: “It is however surprising that C182, being the cysteine with the largest solvent accessible surface area is not bound to MQ in any of the p53-DNA complexes.”

References

- Bykov, V.J., et al. Restoration of the tumor suppressor function to mutant p53 by a low-molecular-weight compound. *Nature medicine* **8**, 282-288 (2002).
- Chen, S., et al. Arsenic Trioxide Rescues Structural p53 Mutations through a Cryptic Allosteric Site. *Cancer cell* **39**, 225-239 (2021).
- Demir, S., et al. Therapeutic targeting of mutant p53 in pediatric acute lymphoblastic leukemia. *Haematologica* **105**, 170-181 (2020).
- Desiraju, G.R. A bond by any other name. *Angew Chem Int Ed Engl* **50**, 52-59 (2011).
- Itoh, Y., et al. N(+)-C-H...O Hydrogen bonds in protein-ligand complexes. *Scientific reports* **9**, 767 (2019).
- Kitayner, M., et al. Structural basis of DNA recognition by p53 tetramers. *Mol. Cell* **22**, 741-753 (2006).
- Kitayner, M., et al. Diversity in DNA recognition by p53 revealed by crystal structures with Hoogsteen base pairs. *Nature structural & molecular biology* **17**, 423-429 (2010).
- Lambert, J.M., et al. PRIMA-1 reactivates mutant p53 by covalent binding to the core domain. *Cancer cell* **15**, 376-388 (2009).
- Lehmann, S., et al. Targeting p53 in vivo: a first-in-human study with p53-targeting compound APR-246 in refractory hematologic malignancies and prostate cancer. *Journal of clinical oncology: official journal of the American Society of Clinical Oncology* **30**, 3633-3639 (2012).
- Panigrahi, S.K. & Desiraju, G.R. Strong and weak hydrogen bonds in the protein-ligand interface. *Proteins* **67**, 128-141 (2007).
- Sallman, D.A., et al. Eprenetapopt (APR-246) and Azacitidine in TP53-Mutant Myelodysplastic Syndromes. *Journal of clinical oncology: official journal of the American Society of Clinical Oncology* **39**, 1584-1594 (2021).
- Wassman, C.D., et al. Computational identification of a transiently open L1/S3 pocket for reactivation of mutant p53. *Nature communications* **4**, 1407 (2013).
- Zhang, Q., Bykov, V.J.N., Wiman, K.G. & Zawacka-Pankau, J. APR-246 reactivates mutant p53 by targeting cysteines 124 and 277. *Cell death & disease* **9**, 439 (2018).

Supplementary Discussion

Does the structure-based study assess the potential rescue of specific cysteines based on the assays by Zhang *et al.* (2018).

The role of specific Cys residues in stabilizing and reactivating mutant p53 has been studied by biochemical and cell-based assays, respectively. Zhang *et al.*⁴ conducted thermal stability measurements of the core domains of wt p53, R273H and R175H proteins where the wt cysteines at positions: 124, 277, 182, 229 and 124/277, were replaced by alanines. The data showed that the replacement of C277 by A277 led to the largest reduction in MQ-mediated thermal stabilization, compared to that of the other Cys to Ala replacements, thus indicating that MQ binding to C277 contributes to thermo-stabilization of the p53 core domain.

The current crystal structures of the isolated core domains of R273H and R273C mutants show that MQ-C277 conjugates form stabilizing contacts with the proteins (**Fig. 4**) and hence likely contribute to increased p53 stability. By contrast, well-defined MQ-C124 conjugates were not resolved in the structures of the two mutants, other than electron density traces (**Table 1A**), suggesting that their contribution to p53 stability in the DNA-free state may not be significant. However, determining the role of MQ-C124 based on thermal stability is complicated as the replacement of C124 by A124 led to different effects in different proteins. The potential role of MQ-C182 on p53 stabilization cannot be confirmed by the structural data because stabilizing contacts are made only with neighboring monomers in the crystal (**Table 1C**). MQ-C229 conjugates in the same structures show high occupancy levels and mutual stabilization between MQ and p53 (**Table 1A, C**), and hence likely contribute to local stabilization of the core domains of these mutants, whereas the data of Zhang *et al.*, showed only a slight but consistent reduction in thermal stability as a result of replacing C229 by A229 in all three proteins. However, effects caused merely by replacing cysteines by alanines could affect the thermal stability data, as exemplified by replacing C124 by A124.

To investigate the role of cysteines on mutant p53 reactivation by APR-246/MQ in tumor cells, Zhang *et al.* used vectors encoding the p53 mutants: R175H, R175H/C124A, R175H/C277A or R175H/C124A/C277A. Based on these data, they proposed that both C124 and C277 were required for MQ-mediated reactivation of R175H in tumor cells⁴. Our structural data of MQ-modified p53-DNA complexes suggest that three MQ-modified residues, C124, C229 and C277, contribute to mutant p53 reactivation. The role of MQ-C124 and MQ-C277 is in accordance with the cell-based assay, whereas the potential role of MQ-C229 was not tested yet. The role of MQ-C124 in transcriptional reactivation of R175H was also proposed by Wassman *et al.*⁵

Specific p53 cancer mutants that may be more amenable to rescue by MQ than others.

On the basis of the available structural data, including the current MQ-modified structures, the previously published structures of wt and mutant p53, and the

published data on mutant p53 reactivation, we propose the potential effects of MQ on other p53 cancer mutants.

The current MQ-modified structures demonstrate that DNA binding of both DNA-contact and structural mutants is supported by a general mechanism mediated by MQ bound to three cysteines (C124, C229 and C277). In addition to R273H bound to both DNA and MQ studied here, it is likely that DNA binding activity of other DNA-contact mutants, including R248Q, R280K and R273C, is restored by the same mechanism where the loss of hydrogen bonds with DNA is compensated for by new MQ-mediated p53-DNA and protein-protein interactions. DNA-binding of these mutants were also shown previously to be restored by PRIMA-1⁶.

Whereas DNA-contact mutants are mostly thermodynamically stable, the structural mutants display various levels of stability and folding ranging from weakly destabilized and locally distorted mutants (e.g. G245S and R249S), to globally denatured mutants (e.g. R175H and R282W)⁷. The current structural data show that the structural mutant R282W displays the common MQ-mediated mechanism, so it is likely that a similar mechanism leads to R175H reactivation. Both mutants were shown to be rescued by APR-246 or PRIMA-1⁴⁻⁶.

Zn ions play a major role in the stability of the p53 core domains and their complexes with DNA^{3,8}. In the structures of p53-DNA tetramers, two central water molecules in each dimer provide a central anchor for an internal hydrogen-bonding network that links the two Zn ions, each coordinated to C176, C238, C242 and H179, thereby supporting the H1 helix and L2 and L3 loops, involved in maintaining the protein-DNA and the protein-protein interfaces within and between p53 dimers^{2,3}. Hence, mutations that directly or indirectly impair the Zn coordination would weaken the stability of the tetrameric p53-DNA complex as shown by certain p53 mutants.

The crystal structures of the cancer mutants G245S and R249S display local structural distortions near the sites of the specific mutations^{9,10,11} where the corresponding mutations, S245 or S249, located at the L3 loop, differently affect the tetrahedral Zn coordination. In G245S, the hydroxyl side chain of S245 interacts with the thiol group of C238, leading to a local distortion of L3 at residues M243-G244. In R249S, the replacement of R249 by S249 results in the loss of a salt bridge and hydrogen bonds between L2 and L3 loops, as well as distorting the Zn coordination by the formation of two alternative conformations of C238¹⁰. DNA binding activity of these and of other mutants was shown to be restored by the presence of second-site mutations presenting alternative stabilizing interactions^{1,9,10}. Hence, such mutants might be rescued by the combined mechanism shown by the current p53-DNA-MQ structures. In the case of G245S, it was also shown that PRIMA-1 led to significant reactivation of PUMA and p21 transcriptional reporters⁵. However, unlike local structural distortions in the Zn regions, direct mutations at the Zn coordination site could lead to global denaturation of p53. Examples of such mutants are C242S which is largely unfolded⁷, not studied yet for APR-246/MQ-mediated reactivation, and C176F being the only exception out of 14 mutants, not showing DNA-binding activity after PRIMA-1 treatment⁶.

The ultimate assessment of MQ-mediated functional rescue of p53 oncogenic mutants will be achieved by structural data combined with quantitative functional assays using a large range of p53 response elements.

Supplementary References

1. Eldar, A., Rozenberg, H., Diskin-Posner, Y., Rohs, R. & Shakked, Z. Structural studies of p53 inactivation by DNA-contact mutations and its rescue by suppressor mutations via alternative protein-DNA interactions. *Nucleic Acids Res* 41, 8748-7859 (2013).
2. Kitayner, M. et al. Diversity in DNA recognition by p53 revealed by crystal structures with Hoogsteen base pairs. *Nat Struct Mol Biol* 17, 423-429 (2010).
3. Kitayner, M. et al. Structural basis of DNA recognition by p53 tetramers. *Mol. Cell* 22, 741-753 (2006).
4. Zhang, Q., Bykov, V.J.N., Wiman, K.G. & Zawacka-Pankau, J. APR-246 reactivates mutant p53 by targeting cysteines 124 and 277. *Cell Death Dis* 9, 439 (2018).
5. Wassman, C.D. et al. Computational identification of a transiently open L1/S3 pocket for reactivation of mutant p53. *Nat Commun* 4, 1407 (2013).
6. Bykov, V.J. et al. Restoration of the tumor suppressor function to mutant p53 by a low-molecular-weight compound. *Nat Med* 8, 282-288 (2002).
7. Bullock, A.N. & Fersht, A.R. Rescuing the function of mutant p53. *Nat Rev Cancer* 1, 68-76 (2001).
8. Cho, Y., Gorina, S., Jeffrey, P.D. & Pavletich, N.P. Crystal structure of a p53 tumor suppressor-DNA complex: understanding tumorigenic mutations. *Science* 265, 346-355 (1994).
9. Joerger, A.C., Ang, H.C. & Fersht, A.R. Structural basis for understanding oncogenic p53 mutations and designing rescue drugs. *Proc Natl Acad Sci U S A* 103, 15056-15061 (2006).
10. Suad, O. et al. Structural basis of restoring sequence-specific DNA binding and transactivation to mutant p53 by suppressor mutations. *J Mol Biol* 385, 249-265 (2009).
11. Chen, S. et al. Arsenic Trioxide Rescues Structural p53 Mutations through a Cryptic Allosteric Site. *Cancer Cell* 39, 225-239 (2021).

REVIEWER COMMENTS

Reviewer #1 (Remarks to the Author):

The authors have added significant literature and extra discussion to address the comments raised and have corrected minor errors. I appreciate these updates, which improve the manuscript to a publishable level. The manuscript will form a useful resource for the p53 community to explore their own follow up investigations.

Reviewer #2 (Remarks to the Author):

The revised manuscript is acceptable for publication.